# Segmentation From Attention: Training-Free Layer Selection and One-Shot Tuning for Segmentation in VLMs

**Mir Rayat Imtiaz Hossain**                                               *rayat137@cs.ubc.ca*
*Department of Computer Science*
*University of British Columbia*
*Vector Institute*

**Mennatullah Siam**                                                       *mennatul@ualberta.ca*
*Department of Computer Science*
*University of British Columbia*

**Leonid Sigal**                                                           *lsigal@cs.ubc.ca*
*Department of Computer Science*
*University of British Columbia*
*Vector Institute*
*CIFAR AI Chair*

**James J. Little**                                                        *little@cs.ubc.ca*
*Department of Computer Science*
*University of British Columbia*

**Reviewed on OpenReview:** *https://openreview.net/forum?id=a5lAwubXro*

## Abstract

Large-scale vision-language models (VLMs), trained on extensive datasets of image-text pairs, exhibit strong multimodal understanding capabilities by implicitly learning associations between textual descriptions and image regions. This emergent ability enables zero-shot object detection and segmentation, using techniques that rely on text-image attention maps, without necessarily training on abundant labeled segmentation datasets. However, performance of such methods depends heavily on prompt engineering and manually selected layers or head choices for the attention layers. In this work, we propose a training-free entropy-based measure, InfoScore, to identify the best image-text attention layers for segmentation, providing a more flexible and scalable solution for training-free open-vocabulary segmentation, reducing the additional burden of hyperparamter search. We empirically show that our training-free selection strategy is superior to naive selection strategies. Additionally, we demonstrate that instead of solely relying on text prompts, fine-tuning the image-text attention layer with a single visual example of each class significantly improves segmentation without the need of additional parameters or decoders. Moreover, we show that our methods and findings are general and can be applied across various vision-language models (VLMs).

## 1 Introduction

In recent years, deep learning research has increasingly focused on foundation models (Bommasani et al., 2021; Li et al., 2024) trained on broad datasets to support generalization across a wide variety of down-

---

*Source code: https://github.com/rayat137/Segmentation-From-Attention.

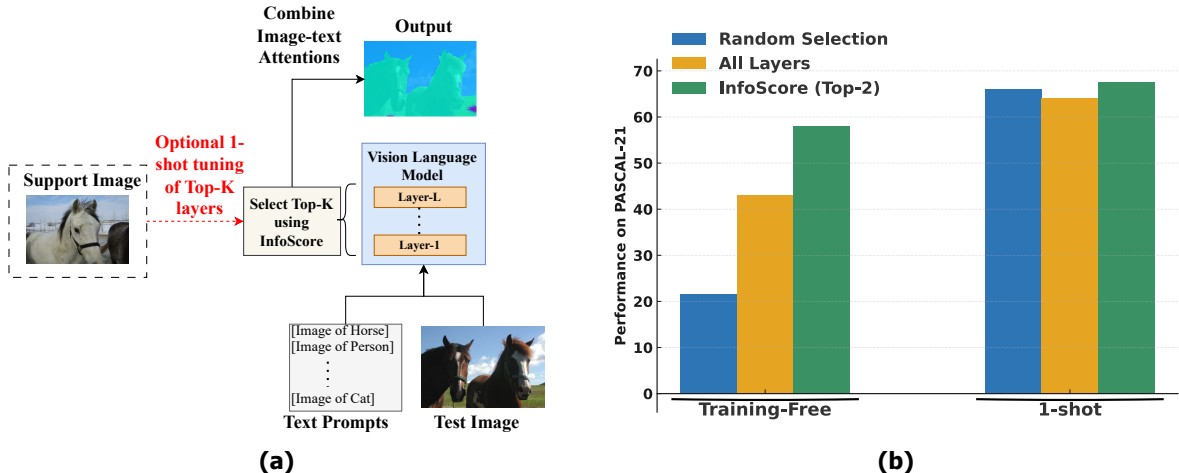

Figure 1: **Overview of Proposed Approach.** **(a)** *Overview:* Given a vision–language model, class-specific text prompts, and a query image, we rank layers using the proposed InfoScore measure, select the top-K attention layers, and aggregate their text-to-image attention maps to generate the final prediction. Optionally, the top-K layers can be fine-tuned with a single support image, enabling both training-free zero-shot and 1-shot settings. **(b)** *Effectiveness of InfoScore:* selecting only the top-2 layers yields significantly superior segmentation compared to random or all-layer aggregation in training-free setting. A single support image further boosts the performance. In this 1-shot setting, InfoScore-based layer selection also outperforms random or all-layer aggregation, although the performance gap is reduced.

stream tasks, typically through self-supervised learning (He et al., 2022; Oquab et al., 2024; Caron et al., 2021) or vision-language modeling (Radford et al., 2021; Xu et al., 2023; Li et al., 2022b; 2023a; Alayrac et al., 2022). Vision-language pre-training, in particular, has seen significant advancements with models like CLIP (Radford et al., 2021) and BLIP (Li et al., 2022b), which leverage image-text contrastive learning or image-text matching. Another family of vision-language models, such as Flamingo (Alayrac et al., 2022) and LLaVA (Liu et al., 2024b), builds upon the capabilities of large language models (LLMs) to support vision-text reasoning tasks, such as visual question answering (VQA) and image captioning. These models are commonly referred to as multi-modal LLMs (MLLMs). Additionally, task-specific foundation models like Segment Anything (SAM) (Kirillov et al., 2023; Ravi et al., 2024) and LLaVA descendants (Chen et al., 2023; Zhang et al., 2024b) focus on pixel-level or region-level understanding, but they typically require complex decoders and extensive datasets with pixel-level annotations to train.

A growing body of research focuses on training-free methods for pixel-level tasks, particularly image segmentation. These approaches leverage vision-language models to enable segmentation in an open-vocabulary setting (Wang et al., 2024a; Hajimiri et al., 2025; Luo et al., 2024; Zhou et al., 2022; Cha et al., 2023; Barsellotti et al., 2024; Wysoczańska et al., 2024b; Lan et al., 2024; Li et al., 2023b; Karazija et al., 2024). However, most existing methods rely on a single type of VLM—often CLIP-based models (Radford et al., 2021)—that form segmentations using similarity between CLIP visual patch features and text features (Wysoczańska et al., 2024b; Wang et al., 2024a; Hajimiri et al., 2025; Zhou et al., 2022; Cha et al., 2023; Lan et al., 2024). Others rely on text-conditioned diffusion models (Barsellotti et al., 2024; Li et al., 2023b; Karazija et al., 2024), typically extracting text-image cross-attention maps from diffusion backbones (Rombach et al., 2022). PNP-OVSS (Luo et al., 2024) extends this idea to VLMs trained with image–text matching (ITM) losses by extracting cross-attention maps, but it does not apply to modern LLM-based VLMs (e.g., LLaVA (Liu et al., 2024b)) that lack ITM heads. Moreover, its layer selection relies on a CLIP-based reward computed using image-level ground-truth class names, while inference additionally requires an external multimodal LLM (e.g., GPT-4o) to identify the set of classes present in each image. Overall, such approaches remain closely tied to specific architectures and external supervision, limiting their generality.

Hence, we propose a model-agnostic framework for open-vocabulary segmentation using vision-language models that operates in both training-free and one-shot fine-tuning modes, as illustrated in Figure 1. Our method is designed to generalize across diverse VLM families, including both cross-attention-based architectures and modern LLM-based vision-language models. We introduce three core components:

1. An entropy-based, training-free measure, **InfoScore**, which ranks and selects the most relevant text-to-image attention layers in an unsupervised manner.

2. A false-positive filtering mechanism that re-weights attention maps using image-text scores derived from the model's own language modeling or image-text matching logits—without relying on external models or annotations.

3. A one-shot tuning strategy that selectively fine-tunes the attention layers identified by InfoScore, without introducing additional decoders or requiring large-scale dense annotations.

In the training-free mode, given class-specific text prompts and a query image, we aggregate the text-to-image attention maps from the top-K layers identified by InfoScore to make final predictions (Figure 1, left). This layer selection yields more effective segmentation than other alternatives (Figure 1, right).

In the one-shot mode, we fine-tune the InfoScore-selected layers together with the corresponding word embeddings using a single support image per class. This helps disambiguate text prompts and better align visual concepts with language. As shown in Figure 1, fine-tuning with even a single visual example yields substantial gains—improving mIoU by nearly 10 points on PASCAL VOC (Everingham et al., 2010).

We evaluate our approach across four benchmark datasets and demonstrate its extensibility across both cross-attention-based vision-language models(*e.g.*, ALBEF (Li et al., 2021), BLIP (Li et al., 2022b)) and modern LLM-based vision-language models (*e.g.*, LLaVA 1.5 (Liu et al., 2024a)). Our training-free method outperforms prior works, and we further show consistent improvements with one-shot fine-tuning.

**Contributions.** In summary, our contributions include:

- A model-agnostic framework for open-vocabulary segmentation that generalizes across both cross-attention-based and LLM-based vision-language models.

- **InfoScore**, an entropy-based measure for training-free, unsupervised selection of informative attention layers.

- A false-positive class filtering mechanism that leverages the model's own image-text scores without requiring external models.

- A one-shot tuning strategy that selectively fine-tunes InfoScore-identified attention layers without introducing additional decoders or requiring large-scale annotations.

- State-of-the-art performance on three out of four benchmark datasets in the training-free setting, and significant additional gains from one-shot fine-tuning, surpassing all existing approaches.

## 2 Related Work

### 2.1 Semantic Segmentation

Semantic segmentation (Arbeláez et al., 2012; 2014; Carreira & Sminchisescu, 2011; Uijlings et al., 2013; Chen et al., 2014; 2017a;b; Long et al., 2015; Cheng et al., 2021; 2022; Hossain et al., 2024) has evolved from early methods based on graph cuts over seeded regions (Boykov & Funka-Lea, 2006; Boykov & Kolmogorov, 2004; Shi & Malik, 2000) and proposal-based mask classification (Arbeláez et al., 2012; 2014; Carreira & Sminchisescu, 2011; Uijlings et al., 2013; Carreira et al., 2012; Dai et al., 2015), to deep learning approaches that frame segmentation as a per-pixel classification task (Chen et al., 2014; 2017a;b; Long et al., 2015).

More recently, models like MaskFormer (Cheng et al., 2021) and Mask2Former (Cheng et al., 2022) have reintroduced the idea of mask classification—predicting a set of binary masks, each paired with a class label—establishing a new paradigm. This formulation underlies many modern methods that incorporate textual and visual prompts. OneFormer (Jain et al., 2023), for example, uses text-conditioned prompts to unify semantic, instance, and panoptic segmentation. SAM (Kirillov et al., 2023), SAM2 (Ravi et al., 2024), and SEEM (Zou et al., 2023) similarly adopt a prompt-driven mask prediction framework, enabling flexible interaction via text, clicks, or bounding boxes.

## 2.2 Open Vocabulary Segmentation

Zero-shot semantic segmentation has been extensively studied to segment unseen, novel classes using text descriptions (Bucher et al., 2019; Zhou et al., 2023). However, these methods do not allow overlap between base and novel classes. Open-vocabulary segmentation addresses this by allowing such overlap and leveraging large-scale vision-language pre-training (Li et al., 2022a; Wu et al., 2024). Training with language data allows the models to handle large, extensible vocabularies, unlike traditional zero-shot settings with predefined base categories. Recent advances in vision-language pre-training (Radford et al., 2021; Li et al., 2022b) have further advanced open-vocabulary image segmentation.

## 2.3 Vision-language Pretraining

Vision-language models (VLMs) can broadly be categorized into two types: (i) models that use cross-attention between image and text modalities for multi-modal reasoning, such as BLIP (Li et al., 2022b) and ALBEF (Li et al., 2021); and (ii) models that integrate Large Language Models (LLMs) by projecting image embeddings into the language space, where they are treated as pseudo-text tokens and processed through self-attention alone (Liu et al., 2024b;a).

BLIP (Li et al., 2022b) jointly trains image and text encoders using a contrastive image-text loss to align the modalities. These aligned representations are then passed to a multi-modal encoder trained with an image-text matching (ITM) loss, further reinforcing cross-modal alignment. ALBEF (Li et al., 2021) adopts a similar two-stage strategy: a contrastive pretraining phase followed by multi-modal fusion using both ITM and masked language modeling (MLM) objectives. In both cases, text-to-image interactions are explicitly handled via cross-attention layers in the transformer architecture.

More recent multi-modal LLMs such as LLaVA (Liu et al., 2024b;a) take a different approach. These models project image embeddings into the language embedding space, where they are concatenated with textual input and processed entirely through causal self-attention. This design eliminates explicit cross-attention layers; instead, the self-attention mechanism implicitly models cross-modal interactions. These models are typically pre-trained and then instruction-tuned using only the next-token prediction objective, similar to large language models.

## 2.4 Prompting VLMs for Training-Free Segmentation

Unlike segmentation-specific models such as SAM (Kirillov et al., 2023) or SEEM (Zou et al., 2024), vision-language models (VLMs) are not explicitly trained for segmentation. Nevertheless, prior work prior work has shown that they can be prompted to produce segmentation maps without additional training (Wang et al., 2024a; Hajimiri et al., 2025; Luo et al., 2024; Zhou et al., 2022; Cha et al., 2023; Barsellotti et al., 2024; Wysoczańska et al., 2024b; Lan et al., 2024; Li et al., 2023b; Karazija et al., 2024). Most of these approaches, however, are limited to a single type or family of VLMs, such as CLIP (Wysoczańska et al., 2024b; Wang et al., 2024a; Hajimiri et al., 2025; Zhou et al., 2022; Cha et al., 2023; Lan et al., 2024) or models with explicit image-text cross-attention layers, including diffusion-based models (Luo et al., 2024; Barsellotti et al., 2024; Li et al., 2023b; Karazija et al., 2024; Rombach et al., 2022). In CLIP-based methods, segmentation is typically derived from the similarity between visual features and text prompts. In contrast, cross-attention-based methods use image-text attention maps as proxies for segmentation. However, (Luo et al., 2024) observed that cross-attention maps tend to over-segment objects and generate numerous false positives, and therefore used GPT-4o to filter out categories not present in the image. In contrast, we observe

that we can refine the heatmaps using an image-text scoring that comes with these VLMs, without the need for any additional models beyond the VLM itself.

Another key insight from approaches that utilize image-text cross-attention maps, or Grad-CAM, for tasks such as segmentation (Luo et al., 2024), visual grounding (He et al., 2024; Li et al., 2021), or visual question answering (Tiong et al., 2022; Li et al., 2021) is that the quality of text-to-image grounding is highly sensitive to the choice of cross-attention layer or head from which attention maps are extracted. Typically, these methods assess layer-wise, task-specific performance on validation sets with ground-truth annotations to select the layer that maximizes performance (Tiong et al., 2022; Li et al., 2021; He et al., 2024), or they use ground-truth class information from validation images (Luo et al., 2024). Both approaches, however, rely on some form of ground truth, which limits their practicality in a truly training-free, open-vocabulary setting where annotations are unavailable, making it challenging to determine which layer's cross-attention map would yield strong segmentation performance. To address this limitation, we propose an entropy-based measure called InfoScore, designed to identify the most suitable layer combinations in a vision-language model (VLM), without annotations.

Furthermore, these methods do not incorporate few-shot or one-shot demonstrations, making them vulnerable to training distributions and vocabularies of the underlying VLMs. In contrast, our approach extends open-vocabulary segmentation by incorporating such demonstrations to guide VLMs in producing segmentation maps. Crucially, our method is applicable to a broad range of VLMs, including recent multimodal large language models (MLLMs) like LLaVA (Liu et al., 2024b).

### 2.5  Quantifying Heatmap and Attention-Map Properties

Regarding determining the desirable properties of heatmaps or attention maps, a line of work has studied the intrinsic properties of saliency maps, aiming to quantify their informativeness or uncertainty. Several approaches (Wang et al., 2010; Kümmerer et al., 2015; Pardyl et al., 2023) adopt information-theoretic frameworks such as entropy, entropy-rate, or information gain to characterize saliency distributions (Wang et al., 2010; Kümmerer et al., 2015) and attention maps (Pardyl et al., 2023).

Beyond information-theoretic formulations, prior work has examined other intrinsic properties of saliency maps, including sparsity, dispersion, and calibration, and proposed quantitative metrics for their evaluation (Gomez et al., 2022; Gupta et al., 2022). A complementary line of research evaluates heatmap quality through perturbation-based and causal criteria, testing whether highlighted regions genuinely influence model predictions. Representative approaches include meaningful and extremal perturbations (Fong & Vedaldi, 2017; Fong et al., 2019), randomized input masking (Petsiuk et al., 2018), and benchmark-based faithfulness and sensitivity analyses (Hooker et al., 2019; Yeh et al., 2019).

Other works have investigated the reliability of saliency methods, showing that some explanations may be insensitive to model parameters or training data (Adebayo et al., 2018; Hedström et al., 2024). In transformer-based models, several studies have further questioned whether attention weights can be directly interpreted as explanations, motivating analyses beyond raw attention visualization (Jain & Wallace, 2019; Serrano & Smith, 2019; Chefer et al., 2021).

While these approaches provide valuable tools for analyzing and comparing heatmaps and attention maps, they are not designed to identify informative layers for segmentation. Our work builds on this literature by leveraging information-theoretic properties of attention maps for training-free layer selection in VLM-based segmentation.

### 2.6  Few-shot Segmentation

Few-shot segmentation has been extensively studied in prior works, focusing primarily on traditional approaches that do not leverage recent advances in vision-language models (Wang et al., 2019; Siam et al., 2019; Shaban et al., 2017; Min et al., 2021). These approaches typically address 1-way or N-way segmentation of novel classes against a background but do not evaluate performance on the base classes seen during pre-training. Recent studies have proposed a generalized few-shot segmentation setting that enables evaluation on both base and novel classes (Tian et al., 2022; Hajimiri et al., 2023; Hossain et al., 2024; Liu et al., 2023).

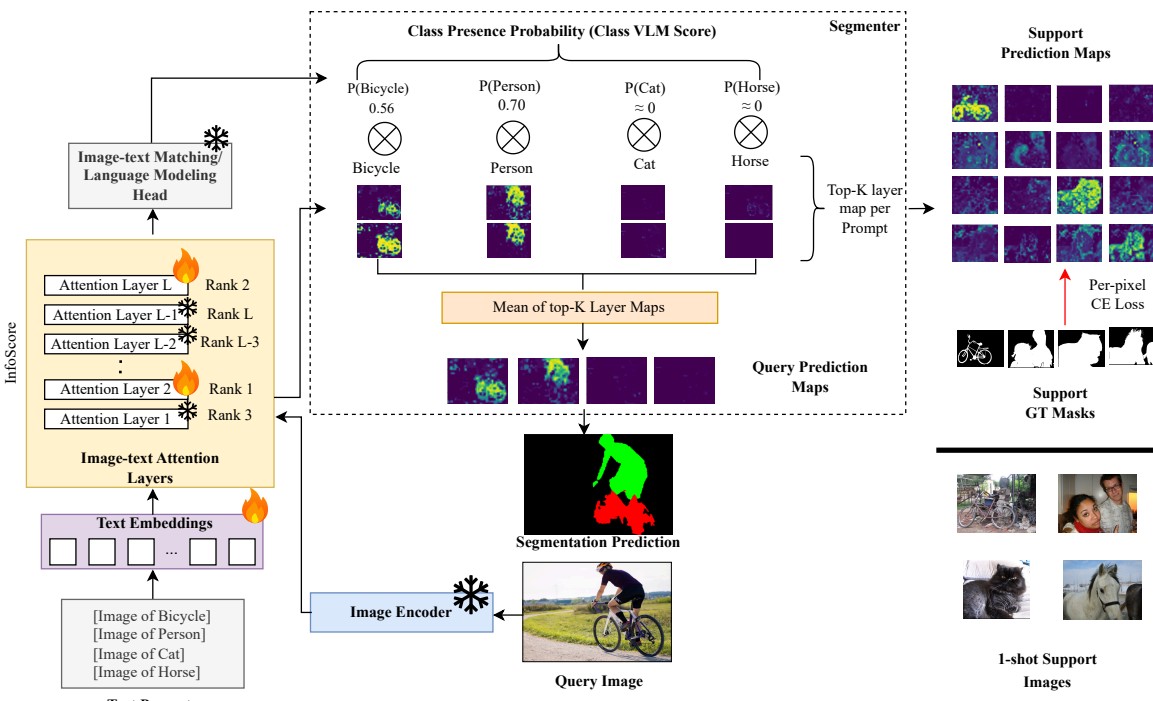

Figure 2: **Model Overview.** Our segmentation framework leverages VLMs trained on image-text pairs, supporting training-free inference and one-shot fine-tuning. For training-free inference, given class names and a query image, we extract text-to-image attention maps from top-$K$ layers (e.g., Layer 2 and Layer $L$), selected via InfoScore (see Sec. 3.2). These maps are re-weighted with class VLM scores to filter irrelevant categories (see Sec. 3.1) and used for prediction. In one-shot fine-tuning, we adjust text embeddings and top-$K$ attention layer parameters (see Sec. 3.4) to further improve the performance.

A recent approach (Hossain et al., 2024) uses a multiscale visual prompting technique to improve segmentation. However, none of generalized approaches utilize advanced vision foundational models, particularly VLMs.

A recent work (Xiao et al., 2024) has explored few-shot adaptation of segment anything (SAM) model (Kirillov et al., 2023). However, SAM is already trained with dense mask annotations over millions of data points. In contrast, we utilize VLMs that were only trained with image-text pairs, and not trained for segmentation. Another approach (Zhu et al., 2024a) explored few-shot segmentation with LLMs and an additional mask decoder by instruction-tuning the LLM on pixel-level annotations to generate 16-point polygons, and training the mask decoder on a large dataset of pixel-level annotations for base categories.

Our approach instead directly mines segmentation maps from VLM text-to-image attention maps, without requiring any external models (e.g., ChatGPT), additional decoders or parameters, and can be applied broadly across different VLMs – themselves trained without any pixel-level annotations.

## 3  Method

Our proposed approach leverages text-to-image attention maps from vision-language models (VLMs). We focus on VLMs that are pre-trained on large-scale image-text pairs without any segmentation supervision, except for the optional use of a single visual exemplar per class (i.e., the one-shot setting). Figure 2 provides an overview of our method, which operates in two modes: a training-free mode and a one-shot fine-tuning

mode. In the training-free setting, we convert the provided class names into natural language prompts and pass them as inputs alongside a query image. We then extract attention maps from the top-$K$ text-to-image attention layers—ranked by our proposed **InfoScore** measure and re-weight them using image-text similarity scores (see Sections 3.1 and 3.2, respectively). In the one-shot setting, we further utilize one labeled visual exemplar per category, including its segmentation mask, to learn an ensemble of attention maps from multiple layers and/or prompts (see Section 3.4).

## 3.1 Training-free Prediction

As shown in Figure 2, during inference we are given a query image $I$ and a class vocabulary $\mathcal{V} = \{c_1, \ldots, c_C\}$. Each class name $c \in \mathcal{V}$ is converted into a natural language prompt of the form `[Image of c]`. The full set of prompts is input to the vision–language model (VLM) together with the image $I$, which is tokenized into a grid of $P \times P$ visual patches.

We consider a cross-attention layer $l$ of the multimodal encoder. Let $T$ be the number of text tokens per prompt. For each class prompt $c \in \mathcal{V}$, the cross-attention operation produces attention maps over the image patches, one per attention head. Since each cross-attention layer contains multiple heads, we compute the element-wise maximum attention score across all heads for each prompt yielding a prompt-specific attention map:

$$\mathcal{A}_c^{(l)} \in \mathbb{R}^{T \times P \times P}.$$

Because a single word in the prompt may be decomposed into multiple sub-word tokens by the tokenizer, we further aggregate across the token dimension to obtain a single spatial attention map per class. Specifically, we compute the mean over the $T$ prompt tokens:

$$\tilde{\mathcal{A}}_c^{(l)} = \frac{1}{T} \sum_{t=1}^{T} \mathcal{A}_c^{(l)}(t)$$

We adopt this aggregation strategy to remain parameter-free and model-agnostic. Among simple parameter-free choices, mean and max pooling are the most common. We choose mean aggregation as it provides a stable estimate across all prompt tokens and avoids over-emphasizing individual tokens, which can occur with max pooling. Stacking these maps over all classes yields a layer-specific attention tensor:

$$\tilde{\mathcal{A}}^{(l)} \in \mathbb{R}^{C \times P \times P}.$$

Normalizing across the $C$ classes using a softmax operation produces a per-patch class probability estimate $\mathcal{M}^{(l)} \in \mathbb{R}^{C \times P \times P}$ .

In cases where we use related words to enhance the class description, similar to previous works (Hajimiri et al., 2025; Wang et al., 2024a), *e.g.*, `man` and `woman` added as related words to the class `person`, we pass them as separate prompts. To compute the heatmap for the corresponding class, we simply take the maximum across the heatmaps of all the related prompts. We call this the **multi-prompt per class setting**. The details of the prompts are provided in Appendix A.5. In the **single-prompt per class setting**, we do not use any related words for a particular class and instead use only the class name as is.

Once per-class attention maps are obtained, we select the top-$K$ cross-attention layers according to the **InfoScore** ranking described in Section 3.2. Let $\mathcal{L}_K \subset \mathcal{L}$ denote the selected layers. Attention maps extracted from transformer-based VLMs are often noisy due to attention sinks, particularly in models that leverage large language models (LLMs) such as LLaVA-1.5 (Liu et al., 2024a), as observed in prior work (Kang et al., 2025; Darcet et al., 2024; Sun et al., 2024). This issue, together with aggregation across multiple prompts, can lead to spurious activations for classes that are not present in the image, leading to false positive predictions.

To mitigate this effect, we introduce a **class VLM score** $\mathcal{S}(c \mid I)$, which estimates the likelihood of class $c$ being present in image $I$. For VLMs trained with an image–text matching (ITM) objective, we observed that given a prompt `[Image of c]`, image–text matching (ITM) scores from ITM head gives an estimate of

whether a class is present in the image or not. Hence, $\mathcal{S}(c \mid I)$ is obtained directly from the ITM scores. For models without an ITM head (e.g., LLaVA-1.5), we instead prompt the model with [Is there class $c$ in the image?  Answer in Yes or No.] and estimate $\mathcal{S}(c \mid I)$ using the probability assigned to the token Yes.

For each selected layer $l \in \mathcal{L}_K$, the class-specific attention maps are weighted by the corresponding class VLM scores,

$$\mathcal{M}_c^{(l)} = \mathcal{S}(c \mid I) \cdot \tilde{\mathcal{A}}^{(l)}, \tag{1}$$

upsampled to the image resolution, and averaged across layers to form an ensemble attention map for each class

$$\mathcal{U}_c = \frac{1}{K} \sum_{l \in \mathcal{L}_K} \mathcal{M}_c^{(l)}. \tag{2}$$

Normalizing across classes at each pixel yields per-pixel class probabilities

$$P_c(x, y) = \frac{\mathcal{U}_c(x, y)}{\sum_{c' \in \mathcal{V}} \mathcal{U}_{c'}(x, y)}, \tag{3}$$

where $x, y$ denotes a pixel location.

While these probabilities can be used directly for segmentation, this approach often underperforms near object boundaries (Luo et al., 2024). Instead, we use them as unary potentials for a Conditional Random Field (CRF) (Lafferty et al., 2001), defined as $U_c(x) = -\log p_c(x)$. In the Fully Connected CRF (Full-CRF) (Krähenbühl & Koltun, 2011), the conditional distribution is given by

$$P(\mathbf{y} \mid I) \propto \exp\left(-E(\mathbf{y} \mid I)\right), \tag{4}$$

with energy

$$E(\mathbf{y} \mid I) = \underbrace{\sum_x U_{y_x}(x)}_{\text{unary term}} + \underbrace{\sum_{x \neq x'} \left[ w_s k_s(x, x') + w_b k_b\big((x, I_x), (x', I_{x'})\big) \right] \mathbf{1}[y_x \neq y_{x'}]}_{\text{pairwise term}}. \tag{5}$$

Here, $k_s$ is a spatial Gaussian kernel and $k_b$ a bilateral (position+color) Gaussian kernel computed from the input image $I$, with $w_s$ and $w_b$ being learnable weights. Since exact inference of the per-pixel probabilities $P(y_x = c \mid I)$ scales quadratically with the number of pixels, it is computationally intractable. FullCRFs approximate these probabilities using mean-field inference with permutohedral lattice filtering, while ConvCRFs (Teichmann & Cipolla, 2018) achieve the same via convolutional Gaussian filtering, enabling efficient GPU-parallel inference. Let $q_c(x)$ denote this approximation of $P(y_x = c \mid I)$. The final segmentation is computed by:

$$\hat{y}(x) = \arg\max_c q_c(x). \tag{6}$$

We outline the procedure of training-free inference for a single image in Algorithm 2.

## 3.2 Ranking Layers Using InfoScore

As noted in previous works (Luo et al., 2024; He et al., 2024; Tiong et al., 2022), and as observed in Figure 5, segmentation performance can vary significantly depending on the layer from which the attention map is extracted. In practice, when performing training-free segmentation in-the-wild, ground-truth dense annotations or information about object presence in the images is unavailable. To address this, we propose an entropy-based measure, **InfoScore**, which can automatically identify the optimal layer(s) for training-free segmentation without requiring ground-truth annotations.

---

**Algorithm 1** Computation of InfoScore

---

**Input:** Unlabeled image set $\mathcal{D}$ with $|\mathcal{D}| = N$; class vocabulary $\mathcal{V}$ with $|\mathcal{V}| = C$; list of candidate layers $\mathcal{L}$.
**Output:** Ranking of layers $\mathcal{R}$ in descending order.

1: **Initialize** for each layer $l \in \mathcal{L}$:
$\qquad S_H^l \leftarrow 0$             ▷ Sum of image entropies
$\qquad S_p^l \leftarrow \mathbf{0} \in \mathbb{R}^C$         ▷ Sum of image-level class distribution $P_\mathbf{I}^l(\hat{\mathbf{Y}}_c)$
$\qquad \mu^l \leftarrow \mathbf{0} \in \mathbb{R}^C$         ▷ Running mean image-level class distribution
$\qquad M_2^l \leftarrow \mathbf{0} \in \mathbb{R}^C$     ▷ Running $2^{\text{nd}}$ moment for variance of image-level class distribution
2: **for** each image $\mathbf{I} \in \mathcal{D}$ **do**
3:      Query the VLM with $(\mathbf{I}, \mathcal{V})$.
4:      **for** each layer $l \in \mathcal{L}$ **do**
5:          Extract image-text attention maps $\mathcal{A}_\mathbf{I}^l$ from layer $l$.
6:          Scale $\mathcal{A}_\mathbf{I}^l$ with class-VLM scores $\mathcal{S}_\mathbf{I}$.
7:          Normalize across classes at each pixel to obtain per-pixel class probabilities $P_{\mathbf{I}_{x,y}}^l(\hat{\mathbf{Y}}_c) \in \mathbb{R}^{C \times H \times W}$.
8:          Spatially average $P_{\mathbf{I}_{x,y}}^l(\hat{\mathbf{Y}}_c)$ to get image-level class distribution $P_\mathbf{I}^l(\hat{\mathbf{Y}}_c) \in \mathbb{R}^C$.
9:          Compute image-level entropy $H^l(\mathbf{I}) = H(P_\mathbf{I}^l(\hat{\mathbf{Y}}_c))$.
10:        $S_H^l \leftarrow S_H^l + H^l(\mathbf{I})$
          $S_p^l \leftarrow S_p^l + P_\mathbf{I}^l(\hat{\mathbf{Y}}_c)$
          $\delta \leftarrow P_\mathbf{I}^l(\hat{\mathbf{Y}}_c) - \mu^l$
11:        $\mu^l \leftarrow \mu^l + \delta/N$                  ▷ Welford update for running mean and variance
          $\delta' \leftarrow P_\mathbf{I}^l(\hat{\mathbf{Y}}_c) - \mu^l$
          $M_2^l \leftarrow M_2^l + \delta \odot \delta'$
12:      **end for**
13: **end for**
14: **for** each layer $l \in \mathcal{L}$ **do**
15:      $H_{Image}^l = S_H^l/N$                 ▷ mean image-level entropy
16:      $P_D^l(\hat{\mathbf{Y}}_c) = S_p^l/N$           ▷ Dataset-level marginal distribution
17:      $H_{\text{Dataset}}^l = H(P_D^l(\hat{\mathbf{Y}}_c))$
18:      $R^l = H_{\text{Dataset}}^l/H_{Image}^l$            ▷ EntropyRatio
19:      $\sigma^l = \sqrt{M_2^l/(N-1)}$           ▷ $\sigma^l \in \mathbb{R}^{|\mathcal{V}|}$
20:      $\text{CoV}^l = \sum_{c \in \mathcal{V}} \frac{\sigma_c^l}{\mu_c^l}$
21:      $S^l = R^l \cdot \text{CoV}^l$               ▷ InfoScore
22: **end for**
23: Rank layers $l \in \mathcal{L}$ in descending order of $S^l$.
24: **return** Ranked list of layers.

---

Given a set of unlabeled images, $\mathcal{D}$, and a vocabulary of class names, the proposed **InfoScore** measure evaluates the predictive uncertainty of text-to-image attention maps from each layer individually. It is designed with the intuition—motivated by the Inception Score (Salimans et al., 2016)—that layers suitable for segmentation should make confident predictions within each image while maintaining diversity across the dataset.

**Design desiderata.** To guide the design of InfoScore in the absence of ground-truth annotations, we define the following desiderata for a layer-selection measure:

1. **Confidence:** Predictions should be confident within each image, favoring sharply peaked class distributions.

2. **Diversity:** Across the dataset, predicted classes should be diverse and not collapse to a single outcome.

---

**Algorithm 2** Segmentation From Attention with InfoScore-Guided Layer Selection

---

**Input:** Unlabeled image set $\mathcal{D}_{\text{rank}}$ for layer scoring; class vocabulary $\mathcal{V}$; candidate layer set $\mathcal{L}$; test image $I$; number of selected layers $K$; CRF module CRF (e.g., Gaussian CRF).

**Output:** Predicted segmentation masks for images in $\mathcal{D}_{\text{seg}}$.

1: **Step 1: Select informative layers via InfoScore**
2: $\mathcal{R} \leftarrow \text{INFOSCORE}(\mathcal{D}_{\text{rank}}, \mathcal{C}, \mathcal{L})$             $\triangleright$ Alg. 1
3: $\mathcal{L}_K \leftarrow \text{TopK}(\mathcal{R}, K)$             $\triangleright$ e.g., $K = 2$, layers $[0, 3]$
4: **Step 2: Segmentation for a single image $I$**
5: Query the VLM with $(I, \mathcal{C})$ to obtain class–VLM scores $\mathcal{S}$ and image–text attention.
6: Initialize ensemble attention map:
    $\mathcal{U} \leftarrow 0 \in \mathbb{R}^{C \times H \times W}$
7: **for** each layer $l \in \mathcal{L}_K$ **do**
8:     Extract cross-attention maps $\mathcal{A}^{(l)}$ for tokens in $\mathcal{C}$.
9:     Reduce over text tokens to obtain attention map per-class (by taking mean)
    $\tilde{\mathcal{A}}^{(l)} \in \mathbb{R}^{C \times P \times P}$.
10:    Rescale per-class attention maps by class–VLM scores $\mathcal{S}$:
    $\mathcal{M}^{(l)} = \mathcal{S} \odot \tilde{\mathcal{A}}^{(l)} \in \mathbb{R}^{C \times P \times P}$
11:    Upsample to image resolution:
    $\hat{\mathcal{M}}^{(l)} \leftarrow \text{Upsample}(\mathcal{M}^{(l)}, H, W)$.
12:    Accumulate into ensemble:
    $\mathcal{U} \leftarrow \mathcal{U} + \hat{\mathcal{M}}^{(l)}$
13: **end for**
14: Average over selected layers:
    $\mathcal{U} \leftarrow \mathcal{U}/K$
15: Normalize across classes at each pixel to obtain a class probability map $\mathcal{P}_c$ (see Equation 3):

16: Apply CRF refinement using the input image:
    $\mathcal{P}_{\text{crf}} \leftarrow \text{CRF}(\mathcal{P}, I)$
17: Obtain final segmentation by per-pixel argmax:
    $y(h, w) = \arg\max_c \mathcal{P}_{\text{crf}}(c, h, w)$
18: **return** $y$

---

   3. **Image sensitivity:** Predictions should vary meaningfully across images, reflecting image-specific content rather than global statistics.

These desiderata directly motivate the three components of InfoScore described next.

The three components of InfoScore are as follows:

- **Mean Image-Level Entropy:** The average per-image entropy of the label marginal distribution summed over pixels. It captures how confident the model is within each image.

- **Dataset-Level Entropy:** The entropy of the class-wise marginal distribution across the entire dataset. It captures the diversity of class predictions across all images.

- **Coefficient of Variation (CoV):** The ratio of the standard deviation to the mean of the predicted class probabilities across the dataset. It quantifies how much the class probabilities vary across different images.

Given the per-pixel probability predictions for an image **I**, we compute the image-level classwise marginal distribution using layer $l$ cross-attention maps, $P_I^l(\hat{\mathbf{Y}}_c)$ as,

$$P_I^l(\hat{\mathbf{Y}}_c) = \frac{1}{hw} \sum_{x=1}^{h} \sum_{y=1}^{w} P_{I_{x,y}}^l(\hat{\mathbf{Y}}_c), \tag{7}$$

where $h, w$ are the height and width of the predicted heatmap (correspondent to $h = P$ and $w = P$ in Section 3.1), $P^l_{I_{x,y}}(\hat{\mathbf{Y}}_c)$ is the probability that the pixel $(x, y)$ belongs to class $c$ which is computed based on the cross-attention maps from layer $l$. Then we compute the image-level entropy $\mathcal{H}^l(\mathbf{I})$ as,

$$\mathcal{H}^l(\mathbf{I}) = -\sum_{c \in V} P^l_I(\hat{\mathbf{Y}}_c) \log P^l_I(\hat{\mathbf{Y}}_c), \tag{8}$$

where $V$ is the vocabulary of the classes to segment. To compute the mean image-level entropy, we take the average over the set of unlabeled images in dataset $\mathcal{D}$,

$$\mathcal{H}^l_{\text{image}}(\mathcal{D}) = \frac{1}{|\mathcal{D}|} \sum_{I \in \mathcal{D}} \mathcal{H}^l(\mathbf{I}). \tag{9}$$

In order to compute the dataset-level entropy over the dataset $\mathcal{D}$, we first calculate the classwise marginal distribution over the entire dataset, $P^l_D(\hat{\mathbf{Y}}_c)$, as,

$$P^l_D(\hat{\mathbf{Y}}_c) = \frac{1}{|\mathcal{D}|} \sum_{\mathbf{I} \in \mathcal{D}} P^l_I(\hat{\mathbf{Y}}_c) \tag{10}$$

The dataset-level entropy $\mathcal{H}_{\text{dataset}}$ is then given by:

$$\mathcal{H}^l_{\text{dataset}}(\mathcal{D}) = -\sum_{c \in V} P^l_D(\hat{\mathbf{Y}}_c) \log P^l_D(\hat{\mathbf{Y}}_c) \tag{11}$$

We define the ratio of dataset-level entropy to image-level entropy as the **Entropy Ratio**:

$$\text{EntropyRatio}(l) = \frac{\mathcal{H}^l_{\text{dataset}}(\mathcal{D})}{\mathcal{H}^l_{\text{image}}(\mathcal{D})} \tag{12}$$

The EntropyRatio can be viewed as an assessment of the classification ability of each text-to-image attention layer. At the image level, desirable layers should make confident predictions, resulting in sharply peaked class distributions for only a few classes within each image and hence low mean image-level entropy. However, minimizing image-level entropy alone can yield trivial solutions, such as always predicting the same class across all images (*e.g.*, a background-biased classifier). At the dataset level, by contrast, we want predictions to cover a diverse set of classes. Under the assumption of independent and identically distributed (i.i.d.) images where all classes are equally likely, high dataset-level entropy indicates that the layer produces a balanced distribution over classes across the dataset, *i.e.*, it is not biased toward a single outcome regardless of input. In essence, a high EntropyRatio identifies layers that balance confident, low-entropy predictions within each image, against diverse, high-entropy predictions across the dataset.

Nevertheless, dataset-level entropy, on its own, does not capture how predictions vary between images. Since it aggregates marginal class probabilities across the dataset, it can be high even if the same set of classes is predicted for every image—producing a globally balanced distribution without meaningful image specificity.

To address this limitation, we incorporate the Coefficient of Variation (CoV) as a multiplicative factor to the EntropyRatio. The CoV measures the relative fluctuation of class probabilities across images, normalized by their mean, thereby favoring layers that produce distinct predictions for different inputs. Therefore, maximizing CoV promotes image-sensitive and diverse predictions, complementing the global balance captured by dataset-level entropy. Figure 3 illustrates this with a toy example of five images and six classifiers, ranging from trivial (Classifier 1: always class 1; Classifier 2: uniform) to ideal (Classifier 6: GT-like). As shown, EntropyRatio alone ranks Classifier 4 above Classifier 5. Although Classifier 4 is confident and predicts two of the three classes, it completely misses class 3. This ranking ignores the meaningful image-to-image

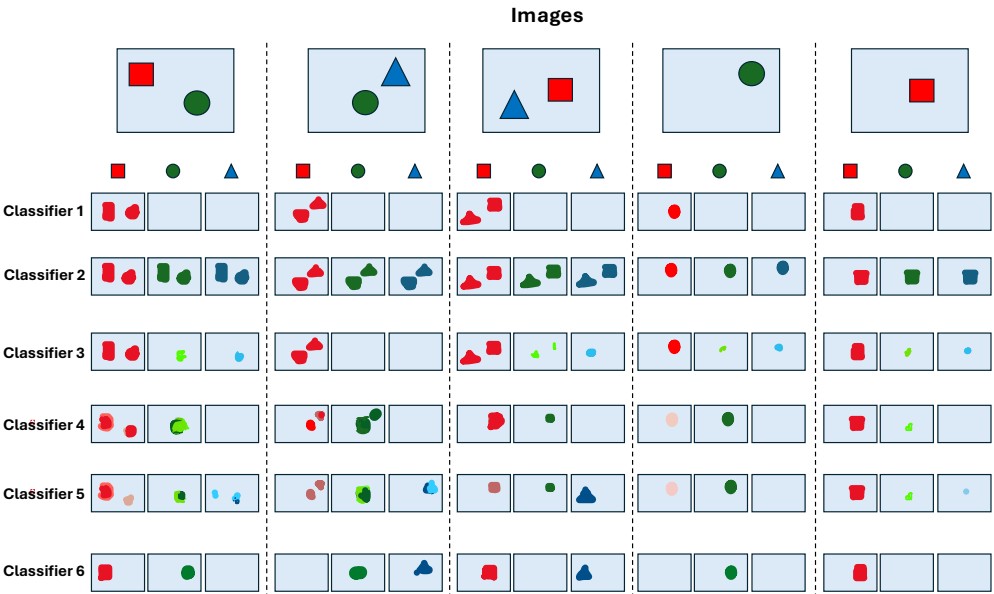

| Classifier | Description | EntropyRatio | | InfoScore | |
|---|---|---|---|---|---|
| | | Value | Rank | Value | Rank |
| Classifier 1 | Dead classifier I – Fully biased towards class 1. | 0.883 | 6th | 0.000 | 5th (tie) |
| Classifier 2 | Dead classifier II – Fully uniform prediction across classes. | 1.000 | 5th | 0.000 | 5th (tie) |
| Classifier 3 | Bad Classifier I – Highly biased towards class 1 with minor variation in prediction. | 1.006 | 4th | 0.294 | 4th |
| Classifier 4 | Bad Classifier II – Sensitive to input but never predicts class 3. | 1.374 | 2nd | 1.600 | 3rd |
| Classifier 5 | Good Classifier – Image sensitive and moderately accurate classifier. | 1.203 | 3rd | 2.299 | 2nd |
| Classifier 6 | Ideal Classifier – GT Like prediction. | 2.546 | 1st | 7.851 | 1st |

Figure 3: **Toy demo of EntropyRatio vs. InfoScore.** We illustrate how InfoScore differs from Entropy-Ratio using synthetic examples. Shown are five images and six classifiers exhibiting different behaviors, from trivial (Classifier 1: always class 1; Classifier 2: uniform) to ideal (Classifier 6: GT-like). EntropyRatio mistakenly ranks Classifier 4 above Classifier 5, despite Classifier 4 never predicting class 3. In contrast, InfoScore—by combining EntropyRatio with the Coefficient of Variation (CoV)—highlights image-sensitive and ideal classifiers. Moreover, while EntropyRatio prefers Classifier 2 over Classifier 1 despite both being non-informative, InfoScore assigns both a value of 0, correctly identifying them as non-informative. Overall, InfoScore balances confidence, diversity, and variability for reliable layer ranking.

variation captured by Classifier 5. InfoScore corrects this by incorporating CoV, correctly ranking Classifier 5 above Classifier 4, emphasizing its image sensitivity. Moreover, while EntropyRatio ranks Classifier 2 (uniform/random prediction) above Classifier 1 (biased to class 1), despite both being non-informative and equally poor, InfoScore assigns them a value of zero, indicating their equally bad performance.

The Coefficient of Variation (CoV) for layer $l$ is computed as:

$$\text{CoV}(l) = \sum_{c \in V} \frac{\sigma_c^l}{\mu_c^l} \tag{13}$$

where $\sigma_c^l$ and $\mu_c^l$ denote the standard deviation and mean, respectively, of the image-level class-wise marginal distribution $P_I^l(\hat{\mathbf{Y}}_c)$ aggregated across the dataset, for each class $c$ at layer $l$.

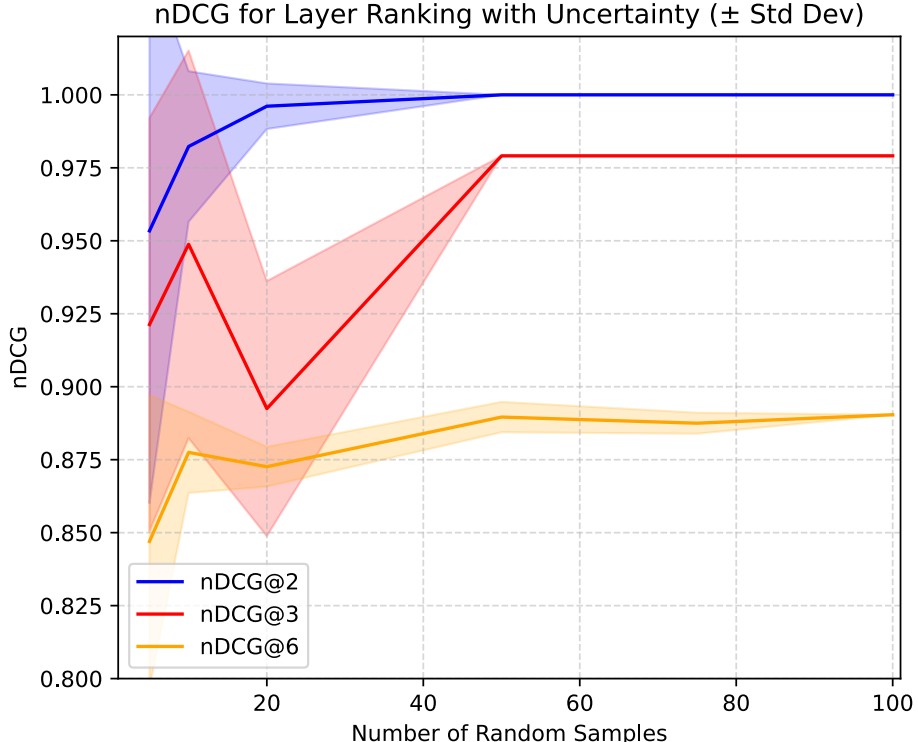

Figure 4: **Convergence of InfoScore.** The InfoScore measure converges for the top-3 ranked layers with as few as 50 random samples and for the top-6 layers with approximately 100 samples. The shaded region indicates uncertainty across repeated trials. The metric nDCG@k (normalized discounted cumulative gain) measures the quality of the ranking for the top-k layers, where a perfect ranking compared to the ground-truth ordering yields a score of 1.0.

Finally, the InfoScore for a layer $l$ is defined as:

$$\text{InfoScore}(l) = \text{EntropyRatio}(l) \times \text{CoV}(l) \tag{14}$$

By maximizing the proposed InfoScore measure, we prioritize layers that balance image-level confidence, dataset-level diversity, and image-specific variability—leading to a more reliable layer selection for training-free segmentation. The procedure for computing InfoScore and ranking layers is detailed in Algorithm 1.

Finally, we note a key assumption underlying InfoScore: object classes are assumed to occupy roughly similar spatial extents within images. In scenarios with consistent size disparities across classes, InfoScore may erroneously favor layers that overlook smaller objects, resulting in suboptimal rankings.

## 3.3 Convergence of InfoScore with Respect to Number of Samples

We do not need to evaluate InfoScore on the full test/validation set of images. In practice, it is sufficient to use a small number of random samples that include images from diverse classes. In Figure 4, we show how the InfoScore converges with respect to the number of random samples. We use the normalized Discounted Cumulative Gain metric (nDCG) (Järvelin & Kekäläinen, 2002) to quantify the ranking quality. The metric nDCG@k measures the quality of the predicted ranking up to the top-k positions, where a score of 1.0 indicates a perfect match with the ground-truth ordering. The nDCG is formally defined as:

$$\text{DCG@k} = \sum_{i=1}^{k} \frac{rel_i}{\log_2(i+1)} \tag{15}$$

where $rel_i$ denotes the relevance of the item at position $i$. The normalized version is given by:

$$\text{nDCG@k} = \frac{\text{DCG@k}}{\text{IDCG@k}} \tag{16}$$

where IDCG@k is the maximum possible DCG achievable with the ideal (ground-truth) ranking.

We compute InfoScore and rank the layers using $n$ randomly selected samples from the PASCAL-21 (Everingham et al., 2010) validation set. For each selection of $n$ random samples, we repeat the experiment five times to account for randomness and report the standard deviation. As observed, as few as 50 random image samples are sufficient for convergence in the top-3 layer ranking (nDCG@3), and approximately 100 samples suffice for top-6 ranking.

### 3.4 One-shot fine-tuning

So far, we have discussed training-free inference using standard VLMs. However, we argue that class names alone are not sufficient to properly ground or segment the corresponding objects. They can be ambiguous or confusing without the appropriate context. For example, the COCO-Obj dataset includes a class name `tie`, which can be ambiguous unless additional context is provided to indicate it refers to a piece of clothing. Therefore, we hypothesize that fine-tuning certain parameters of a VLM pretrained on an image-text retrieval task, using one visual example with dense annotations, can improve the overall segmentation quality.

In few-shot segmentation literature, whether performing 1-way or k-way classification, classes are typically divided into non-overlapping splits of base and novel categories. Models are first trained on base categories with a large number of annotated examples to learn how to perform segmentation, and are then subsequently adapted to novel classes in a few-shot setting. In contrast, our approach does not involve base classes, as the models we use were never explicitly trained for segmentation, meaning all classes are novel. To ensure fairness, only one novel category is present in each selected image. However, in practice, certain novel classes may co-occur. If a one-shot example image for one class contains other classes, we assign them to either the background (if the dataset includes a background category) or to an ignored class.

During one-shot tuning, we minimize a per-pixel cross-entropy loss between the predicted segmentation map and the ground-truth annotations of the support image, updating only the word embeddings and the parameters of the top-$K$ ranked attention layers.

It is worth noting that we do not introduce any additional parameters or decoders for fine-tuning, merely encouraging the attention maps to align with the support image ground truth. Moreover, optimizing only a subset of parameters leads to more data-efficient training and limits overfitting, which is necessary given that we are only leveraging a single (or very few) labeled examples for training.

## 4 Experimental Results

### 4.1 Experimental Setup

**Datasets.** We evaluate our method on four commonly used segmentation datasets: PASCAL-21 (Everingham et al., 2010), COCO-Obj (Lin et al., 2014), COCO-Stuffs-171 (Caesar et al., 2018) and ADE-20K (Zhou et al., 2017). PASCAL-21 and COCO-Obj include a background category, while COCO-Stuffs-171 and ADE-20K do not, but are the most challenging due to the inclusion of `stuff` classes. Following previous works (Wang et al., 2024a; Hajimiri et al., 2025), the background class is represented as a list of possible background categories. Details about the class names used for single and multi-prompt setting are provided in the supplementary. For one-shot evaluation, we conduct five separate runs and average their mIoU.

**Implementation Details.** Implementation details regarding the model weights used for BLIP (Li et al., 2022b), LLaVA (Liu et al., 2024a) and ALBEF (Li et al., 2021) are provided in the supplementary. We report results on images resized to a maximum side length of 512 for BLIP and ALBEF and 448 for LLaVA. By default, we use top-2 layers for all datasets and models except for ADE-20K where top-1 performs slightly better. For ConvCRF, we use a kernel size of $15 \times 15$ and perform 20 iterations per image. We follow (Teichmann & Cipolla, 2018) for other hyper-parameters of ConvCRF.

| Method | VLM | PASCAL-21 | COCO-Obj | COCO-171 | ADE-20K |
|---|---|---|---|---|---|
| *Weakly Supervised Training* | | | | | |
| GroupVIT (Xu et al., 2022) | CLIP | 52.3 | 27.5 | 15.3 | 10.4 |
| ReCo (Cha et al., 2023) | CLIP | 51.2 | 30.4 | 19.6 | 11.2 |
| GroundEverything (Bousselham et al., 2024) | MetaCLIP | 46.8 | - | - | 17.1 |
| SAM-CLIP (Wang et al., 2024b) | CLIP | 60.6 | 31.5 | - | 17.1 |
| Clip-DINOiser (Wysoczańska et al., 2024a) | CLIP | 62.1 | 34.8 | 24.6 | 20.0 |
| *Training Free* | | | | | |
| PNP-OVSS (Luo et al., 2024) | BLIP | 51.3 | 36.2 | 17.9 | 14.2 |
| SCLIP (Wang et al., 2024a) | CLIP | 61.7 | 33.2 | 23.9 | 17.8 |
| ProxyCLIP (Lan et al., 2024) | CLIP+DINO | 61.3 | 37.5 | 26.5 | 20.2 |
| NACLIP (Hajimiri et al., 2025) | CLIP | **64.1** | 36.2 | 25.7 | 19.1 |
| **Ours** | BLIP | 60.2 | **42.8** | **28.1** | **22.8** |
| *One-shot Supervision* (Average Across 5 runs) | | | | | |
| **Ours** | BLIP | **70.1 $\pm$ 0.89** | **45.3 $\pm$ 1.32** | **29.3 $\pm$ 0.05** | **24.5$\pm$ 0.13** |

Table 1: **Comparison of our approach with state-of-the-art methods** for weakly-supervised, training-free, and one-shot supervised open-vocabulary semantic segmentation (OVSS). The results reported on this table are using **BLIP top-2 layers (selected by InfoScore)** in **multi-prompt per class setting**. Best results for training-free setting and the 1-shot results are in **bold**.

For one-shot supervision we use a batch size of two. We fine-tune all models (BLIP, ALBEF, LLaVA) on a single A40 GPU with 48GB memory. We use a learning rate of $2 \times 10^{-4}$ for PASCAL-21 and COCO-Obj, and $5 \times 10^{-5}$ for COCO-171 and ADE-20K. We train COCO-Obj for two epochs, COCO-171 and ADE-20K for three epochs and PASCAL-21 for five epochs. Further details are provided in **supplementary materials.**

Source code for our approach is publicly available at https://github.com/rayat137/Segmentation-From-Attention.

## 4.2 Quantitative Results

### 4.2.1 Comparison to the State of the Art

We compare against the state-of-the-art methods in open vocabulary segmentation. We mainly compare against training-free methods (Luo et al., 2024; Wang et al., 2024a; Hajimiri et al., 2025) and the ones that further fine-tune pre-trained VLMs on large-scale image-text pairs or pseudo annotations through weak supervision (Xu et al., 2022; Cha et al., 2023). The choice for open vocabulary segmentation setup is motivated for the sake of fair comparison, since previous training-free methods had access to aligned vision-language data similar to our approach in its training-free mode. In Table 1, it is shown that our training-free open-vocabulary method outperforms the state-of-the-art ProxyCLIP (Lan et al., 2024) on three challenging benchmark datasets COCO-Obj, COCO-171 and ADE-20K by 5.3%, 1.6% and 2.6%, respectively, while achieving competitive performance on PASCAL-21. Additionally, when provided with a single visual example and only a few iterations of fine-tuning on a small number of parameters, the segmentation perfromance significantly improves across all four datasets, with an improvement of 9.9%, 2.5%, 1.2% and 1.7% for PASCAL-21, COCO-Obj, Coco-171 and ADE-20K respectively. This emphasizes the importance of visual aids—even a single visual example can reduce ambiguities that may arise from text prompts alone. Examples include the `stuff` classes in COCO-171 that are inherently ambiguous (e.g., `solid` or `structural`). The results in this table are reported using BLIP top-2 layers in multi-prompt setting. Single vs. Multi-prompt performance and extensibility on other VLMs is discussed next.

| Method | Base Training | S0 | S1 | S2 | S3 | Average |
|---|---|---|---|---|---|---|
| *Few-shot segmentation approaches strictly separating base and novel classes* | | | | | | |
| LLaFS (CVPR 2024) (Zhu et al., 2024a) | ✓ | 74.2 | **78.8** | 72.3 | 68.5 | 73.5 |
| HMNet (NeurIPS 2024) (Xu et al., 2024) | ✓ | 72.2 | 75.4 | 70.0 | 63.9 | 70.4 |
| AMFormer (NeurIPS 2023) (Wang et al., 2023) | ✓ | 71.3 | 76.7 | 70.7 | 63.9 | 70.7 |
| HDMNet (CVPR 2023) (Peng et al., 2023) | ✓ | 71.0 | 75.4 | 68.9 | 62.0 | 69.3 |
| ACBC (CVPR 2024) (Zhu et al., 2024b) | ✓ | 73.0 | 76.0 | 69.7 | 69.2 | 72.0 |
| OCNet (ICCV 2025) (Wen et al., 2025) | ✓ | 73.5 | 75.9 | 71.1 | 64.9 | 71.4 |
| **Ours** | × | 70.4 | 73.7 | **75.4** | 72.2 | 72.9 |
| **Ours w/ base training** | ✓ | **74.4** | 77.7 | 75.2 | **73.5** | **75.2** |
| *Categories in training cover categories in testing* | | | | | | |
| GraphFSS* (NeurIPS 2024) (Zhang et al., 2024a) | ✓ | – | – | – | – | 72.1 |
| Matcher* (ICLR 2024) (Liu et al., 2024c) | ✓ | – | – | – | – | 68.1 |

Table 2: **Comparison with SOTA 1-shot segmentation approaches on Pascal-$5^i$.** Results are reported under the standard 1-shot setting using four disjoint splits (S0–S3), where each split treats a different set of five classes as novel and the remaining classes as base. Our method without base-class training remains competitive with prior work, achieving the second-best average performance after (Zhu et al., 2024a). Incorporating base-class pretraining yields consistent improvements across all splits and achieves state-of-the-art average performance. * denotes methods that do not strictly enforce base–novel class separation. Best results are shown in **bold**.

### 4.2.2 Comparison to 1-shot Segmentation Approaches

As mentioned in section 3.4, in the conventional few-shot segmentation protocol, semantic classes are partitioned into disjoint base and novel sets across multiple dataset splits. Models are first trained on base classes using abundant segmentation annotations and are subsequently adapted to novel classes using only a single annotated support example per class. Our method on the other hand does not involve any pre-training on base classes with abundant data.

In Table 2, we compare our approach against a range of recent state-of-the-art few-shot segmentation methods that strictly follow the standard few-shot protocol, which evaluates performance over four disjoint splits (S0–S3), each defining a different set of novel classes. As observed, our method achieves competitive performance even without any base-class segmentation training, placing it in a substantially more challenging and annotation-efficient setting.

For completeness, we also report results when our model is trained under the standard few-shot protocol, including base-class training with segmentation annotations. Under this fair setting (strictly following few-shot segmentation protocol), our approach achieves state-of-the-art performance, outperforming several prior methods.

We further distinguish methods that do not enforce strict base–novel separation and allow novel-class exposure during base training, and report their results separately for clarity. Overall, these results demonstrate that our comparisons are fair and comprehensive, and that the proposed approach remains effective across both training-free and standard few-shot settings.

### 4.2.3 Layer-wise mIoU and Layer Ranking using InfoScore

Figure 5 presents the training-free segmentation performance of BLIP (in the **single-prompt per class setting**) on the Pascal-21 validation set when each layer is independently used for segmentation, *i.e.*, layer-wise mIoU (in red). As shown, segmentation performance varies substantially across layers; the best-performing layer achieving an mIoU of 55.6% and the lowest-performing layer achieving an mIoU of only 6.3%.

We also report the InfoScore (in blue) for each of the 12 layers of BLIP along with their corresponding rankings. As observed, the proposed InfoScore effectively identifies the top-performing layers on Pascal-21. In most cases, the InfoScore ranking closely aligns with the mIoU ranking, with only minor deviations. This demonstrates the importance of selecting or extracting attention maps from layers that exhibit strong grounding between text and image, which in turn leads to improved segmentation performance.

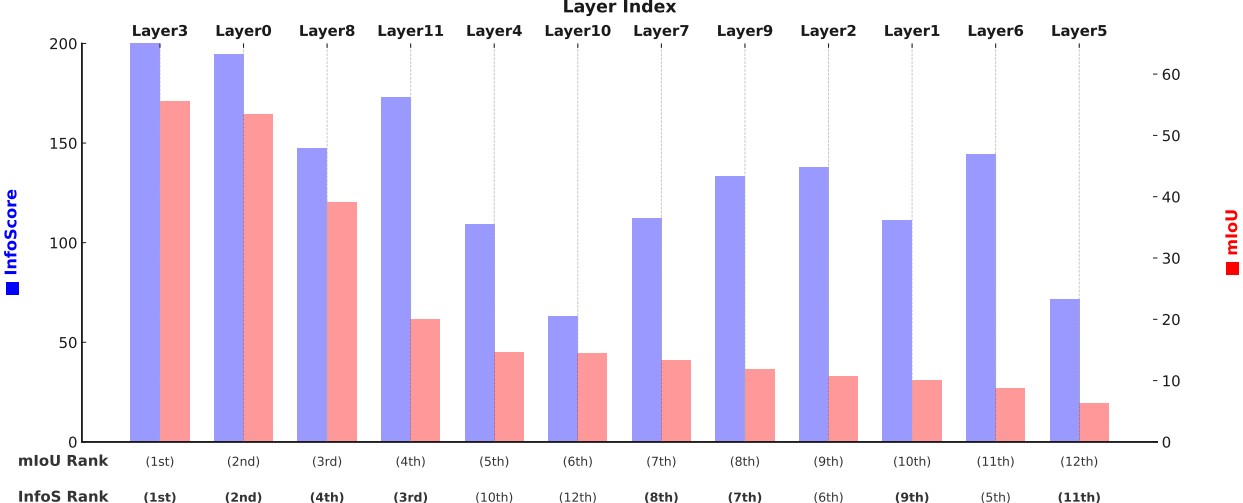

Figure 5: **Illustration of the InfoScore measure on BLIP.** The mIoU Rank reflects the descending order of mIoU values (in red) derived from cross-attention maps for each standalone layer (labeled Layer$N$, top) on the PASCAL VOC 2012 validation set (1449 images), compared to the predicted InfoS Rank (bottom) based on our InfoScore measure (in blue) requiring no annotations. Most InfoScore rankings align with mIoU Rankings, with minor displacements of $\pm 2$ positions highlighted in bold, except for four layers. Empirically, the top-1 and top-2 layers are correctly identified and consistently deliver better performance across four datasets and three VLMs. Results shown here are in **single-prompt per class setting**.

| Layer Selection | PASCAL-21 | COCO-Obj | COCO-171 |
|---|---|---|---|
| *Training Free* | | | |
| All layers (12 layers) | 42.8 | 35.9 | 25.1 |
| Random (1 layer) | 21.5 | 15.6 | 11.0 |
| Naive (First 2 layers) | 48.2 | 37.2 | 25.5 |
| Naive (Last 2 layers) | 19.6 | 10.0 | 6.8 |
| InfoScore (Top-1) | 55.6 | 39.0 | 25.9 |
| **InfoScore (Top-2)** | **58.0** | **42.6** | **28.3** |
| InfoScore (Top-3) | 51.3 | 38.0 | 27.4 |
| InfoScore (Top-6) | 49.2 | 37.0 | 25.2 |
| *One-shot Supervision* | | | |
| InfoScore (Top-1) | 66.8 | 44.3 | 26.6 |
| **InfoScore (Top-2)** | **67.5** | **45.4** | **28.9** |

Table 3: **Analysis of InfoScore-based layer selection in BLIP.** Using an ensemble of attention maps from the top-1, top-2, or top-3 layers ranked by the InfoScore measure outperforms ensembling all layers, random selection, or naive selection. Results are reported under the **single-prompt per class setting**; best results are shown in **bold**.

In Table 3, we further demonstrate the benefits and effectiveness of our layer selection strategy using InfoScore compared to naive or random selection strategies on BLIP. The results show that using an ensemble of the top attention maps, ranked by the proposed InfoScore, significantly outperforms simple aggregation strategies such as averaging attention maps from all layers, as well as `naive` and `random` baselines across all three datasets: Pascal-21, COCO-Obj, and COCO-171.

| Method | PASCAL-21 | | COCO-Obj | | COCO-171 | |
|---|---|---|---|---|---|---|
| | Training-Free | One-Shot | Training-Free | One-Shot | Training-Free | One-Shot |
| *Single* | 58.0 | 67.5 | 42.6 | **45.4** | **28.3** | 28.9 |
| *Multiple* | **60.2** | **70.1** | **42.8** | 45.3 | 28.1 | **29.3** |

Table 4: **Ablation of single vs. multiple prompts with related words.** Multiple prompts consistently improve segmentation accuracy on PASCAL-21 and COCO-171, while their effect on COCO-Obj is less pronounced. Best results are in **bold**.

| Method | Layers | PASCAL-21 | | COCO-Obj | |
|---|---|---|---|---|---|
| | | *Training-Free* | *One-Shot* | *Training-Free* | *One-Shot* |
| **Cross-Attention Based VLMs** | | | | | |
| BLIP | Top-1 | 55.6 | 66.8 | 39.0 | 44.3 |
| BLIP | Top-2 | **58.0** | **67.5** | **42.6** | **45.4** |
| ALBEF | Top-1 | 37.9 | 65.1 | 28.7 | 37.7 |
| ALBEF | Top-2 | 43.2 | 65.5 | 31.7 | 38.7 |
| **LLM-Based VLMs** | | | | | |
| LLaVA-1.5-7B | Top-1 | 50.1 | 63.7 | 31.3 | 41.0 |
| LLaVA-1.5-7B | Top-2 | 51.6 | 64.2 | 32.3 | 42.2 |

Table 5: **Segmentation performance across different vision-language models (VLMs).** Fine-tuning with a single example significantly improves performance for BLIP (Li et al., 2022b), ALBEF (Li et al., 2021), and LLaVA-1.5-7B (Liu et al., 2024a). Results shown are for the **single-prompt per class setting**. Best results are **bolded**.

Additionally, the results show that using the top-2 layers consistently outperforms using only the top-1 layer. Performance tends to saturate at the top-3 layers and declines when more layers are added (*e.g.*, top-6). As noted before, our ranking also surpasses other naive strategies, including selecting the first or last two layers, aggregating attention maps across all layers, or randomly selecting a layer. For the `Random` baseline, we evaluate every layer once (12 runs), compute the mIoU for each layer individually, and report the average (expectation) across all runs, mimicking unbiased uniform sampling.

## 4.3 Single versus Multi-prompt setting

In Table 4 we compare the performance difference between using a single prompt corresponding to each class name against the use of multiple prompts of related words to that specific class name. It shows that multiple prompts either provide better or on-par performance to the single prompt across both training free and one-shot supervision modes.

### 4.3.1 Extensibility Across VLMs

Table 5 demonstrates the extensibility and practicality of our approach, which relies on our proposed In-foScore for layer selection, across diverse vision-language models (VLMs), including recent LLM-based architectures such as LLaVA-1.5. Across all three VLMs, incorporating a single visual example consistently leads to substantial performance gains without introducing additional parameters or classifiers. Among the evaluated models, BLIP achieves the best overall performance on both the PASCAL-21 and COCO-Obj benchmarks, outperforming ALBEF and LLaVA-1.5 in both the training-free and one-shot supervision settings. Additionally, using the Top-2 layers selected by the InfoScore measure yields better performance than using only the Top-1 layer in the training-free scenario across all VLMs, although this advantage diminishes after fine-tuning.

| Layer Selection | BLIP (12 layers) | ALBEF (6 layers) | LLaVA-1.5-7B (32 layers) |
|---|---|---|---|
| All layers | 42.8 | 34.3 | 49.7 |
| Random (1 layer) | 21.5 | 20.6 | 39.3 |
| Naive (First 2 layers) | 48.2 | 38.4 | 26.3 |
| Naive (Last 2 layers) | 19.6 | 11.0 | 34.4 |
| InfoScore (Top-1) | 55.6 | 37.9 | 50.1 |
| **InfoScore (Top-2)** | **58.0** | **43.2** | **51.6** |
| InfoScore (Top-3) | 51.3 | 39.6 | 49.8 |
| InfoScore (Top-6) | 49.2 | 34.3 | 49.5 |

Table 6: **Extensibility and effectiveness of InfoScore-based layer selection in the training-free setting on PASCAL-21.** Across three distinct architectures—BLIP, ALBEF, and LLaVA-1.5-7B—InfoScore consistently identifies the most informative layers. In all cases, selecting the **Top-2 layers** ranked by InfoScore achieves the best mIoU, outperforming random selection, naive heuristics, and averaging all layers. Results shown are in **single-prompt per class setting**.

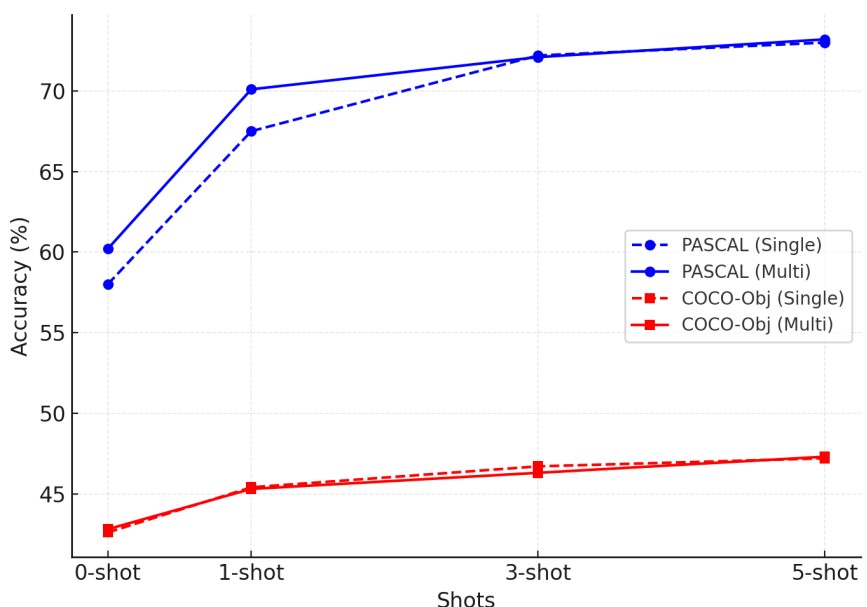

Figure 6: **Performance analysis with increased number of shots for single and multi-prompt setting.** Our model scales as the number of shots increases from 1 to 5 on both COCO-Obj and PASCAL-21 datasets. **Dashed lines** denote the **single-prompt per class setting**, and **solid lines** denote the **multi-prompt per class setting**. The benefit of using multiple prompts diminishes with more shots.

### 4.3.2 Effectiveness of InfoScore-based layer selection across VLMs

To demonstrate that our InfoScore-based layer selection strategy generalizes across VLMs beyond BLIP, Table 6 reports results on ALBEF and LLaVA-1.5-7B in addition to BLIP. Across all three VLMs, the **Top-2 layers selected by InfoScore consistently achieve the highest mIoU**, outperforming random selection, naive heuristics, and averaging attention maps over all layers.

Since LLaVA-1.5-7B has 32 layers, a full sweep would be computationally expensive. Instead, we perform 5 independent runs: in each run, we uniformly sample one layer, compute its mIoU, and report the average across the 5 runs. The sampled layers in our experiments were `Layer 7`, `Layer 0`, `Layer 16`, `Layer 14` and `Layer 27`. For ALBEF, we follow the same approach as BLIP (see Sec. 4.2.3).

| Percent of Seen Classes | Unseen mIoU | | | | |
|---|---|---|---|---|---|
| | **S1** | **S2** | **S3** | **S4** | **Ave** |
| 0% (zero-shot) | 63.6 | 57.2 | **59.9** | 59.5 | 60.1 |
| 25% (one-shot) | **64.8** | **60.0** | 59.0 | **62.8** | **61.7** |

Table 7: **Comparing fine-tuning with 1-shot learning on 25% of the classes (75% unseen) against zero-shot performance on the unseen classes in PASCAL-21.** The seen and unseen classes are divided into four non-overlapping splits. Fine-tuning with just one visual example per class for 25% of the classes improves performance on the 75% unseen categories. Results shown are in **multi-prompt per class setting**.

### 4.3.3 Extensibility Beyond One-Shot

We further examine the scalability of our method beyond the one-shot setting. Figure 6 compares the 3-shot and 5-shot performance against the 1-shot baseline in both single-prompt and multi-prompt settings on the PASCAL-21 and COCO-Obj datasets. All results are averaged over five runs to ensure stability. As expected, increasing the number of shots leads to consistent improvements of approximately 1–2% on both datasets. Notably, the performance gap between single-prompt and multi-prompt settings narrows as the number of shots increases, suggesting that our method does not strongly depend on prompt engineering when a few visual demonstrations are provided beyond the one-shot scenario.

### 4.3.4 One-shot Tuning with Partial Vocabulary

Table 7 illustrates the impact of one-shot fine-tuning on a small subset of the vocabulary and how this adaptation transfers to unseen categories. Specifically, we fine-tune the model using a single visual example for only 25% of the total classes, leaving the remaining 75% entirely unseen during fine-tuning. The classes are divided into four non-overlapping splits; for each split, we fine-tune on one-shot examples from 25% of the classes and evaluate performance on the remaining 75%.

The results show that even with such minimal supervision—just one example per class for a quarter of the vocabulary—the fine-tuned model significantly outperforms the zero-shot baseline on the unseen categories. This demonstrates the model's ability to generalize from a small, partially labeled subset to a much larger set of unseen classes, likely benefiting from the shared semantic structure among categories.

Overall, this finding underscores the practicality of leveraging limited labeled data to boost performance in settings where many classes remain unlabeled, offering a scalable solution for resource-constrained scenarios.

## 4.4 Ablation Study

### 4.4.1 Impact of Image-Text Scoring

Table 8 analyzes the impact of re-weighting the image-text attention maps using the class VLM scores already computed by the VLMs. Incorporating this re-weighting in our design leads to significant improvements in segmentation performance for both training-free and one-shot settings. While the benefit of using class VLM scores is slightly reduced in the one-shot setting, it remains crucial for achieving strong training-free performance in both cross-attention-based VLMs like BLIP and LLM-based VLMs like LLaVA.

### 4.4.2 Ablation with Individual Components of InfoScore

In Table 9, we compare the layer rankings of the BLIP model obtained using individual components of the InfoScore measure on the PASCAL-21 and ADE-20K datasets. As shown, relying solely on either Image Entropy or Dataset Entropy leads to suboptimal layer selection and consequently poor segmentation performance.

| Method | Class VLM Score | PASCAL-21 | | COCO-Obj | |
|---|---|---|---|---|---|
| | | *Training-Free* | *One-Shot* | *Training-Free* | *One-Shot* |
| BLIP | × | 25.1 | 62.4 | 26.0 | 38.7 |
| BLIP | ✓ | **58.0** | **67.5** | **42.6** | **45.4** |
| LLaVA-1.5-7B | × | 36.8 | 55.0 | 20.1 | 34.4 |
| LLaVA-1.5-7B | ✓ | 51.6 | 63.7 | 32.3 | 42.1 |

Table 8: **Ablation on the Importance of Image-Text Scoring.** Re-weighting the image-text attention maps using the class VLM scores significantly enhances performance in the training-free scenario and provides additional gains in the one-shot setting. The benefit of this scoring is more pronounced without fine-tuning, though it still contributes to improved results with one-shot supervision. Results are reported for the **single-prompt per class setting**. Best performances are **bolded**.

| Ranker | PASCAL-21 | | ADE-20K | |
|---|---|---|---|---|
| | nDCG@2 | mIoU@2 | nDCG@2 | mIoU@2 |
| Image Entropy | 0.43 | 9.2 | 0.57 | 8.9 |
| Dataset Entropy | 0.67 | 19.1 | 0.57 | 8.9 |
| EntropyRatio | **1.00** | **58.1** | 0.93 | 16.8 |
| **InfoScore** | **1.00** | **58.1** | **0.98** | **22.8** |

Table 9: **Ablation of InfoScore Components for Layer Ranking of the BLIP Model on PASCAL-21 and ADE-20K.** We report nDCG@2 and mIoU@2 for each ranker on both datasets, which respectively indicate the ranking quality and the training-free segmentation performance (mIoU) of the top-2 layers. Results are shown for **single-prompt per class setting**.

Using only Image Entropy favors layers biased toward a few dominant classes, while Dataset Entropy favors layers predicting many classes per image but with low confidence. Both lead to poor segmentation. Entropy-Ratio (Sec. 3.2) balances these effects by promoting layers with confident per-image predictions and global class diversity, yielding strong results on PASCAL-21. However, it overlooks image-to-image variation and can mis-rank layers when global balance masks low variability, as seen on ADE-20K.

InfoScore resolves this by multiplying EntropyRatio with the Coefficient of Variation (CoV), explicitly modeling prediction variability across images. As Table 9 shows, this consistently produces the best rankings and significantly improves training-free segmentation, especially on ADE-20K.

| Fine-Tuned Parameters | PASCAL-21 | COCO-Obj |
|---|---|---|
| Top-2 Cross-Attention Layers Only | 65.0 | 44.4 |
| Word Embeddings Only | 63.2 | 43.4 |
| ├─ + Random-2 Cross-Attention Layers | 66.1 | 42.4 |
| ├─ + Top-2 Cross-Attention Layers | **67.5** | **45.4** |
| ├─ + All Cross-Attention Layers | 64.1 | 40.6 |
| └─ + All VLM Parameters | 60.9 | 41.9 |

Table 10: **Fine-tuning Strategies.** The top section shows the standalone impact of fine-tuning the top-2 cross-attention layers. The bottom section presents the progressive addition of parameters starting from fine-tuning word embeddings. Fine-tuning both the word embeddings and the top-2 layers yields the best performance, while further increasing the number of tunable parameters leads to reduced performance. Results are reported in the **single-prompt per class setting**. Best results are **bolded**.

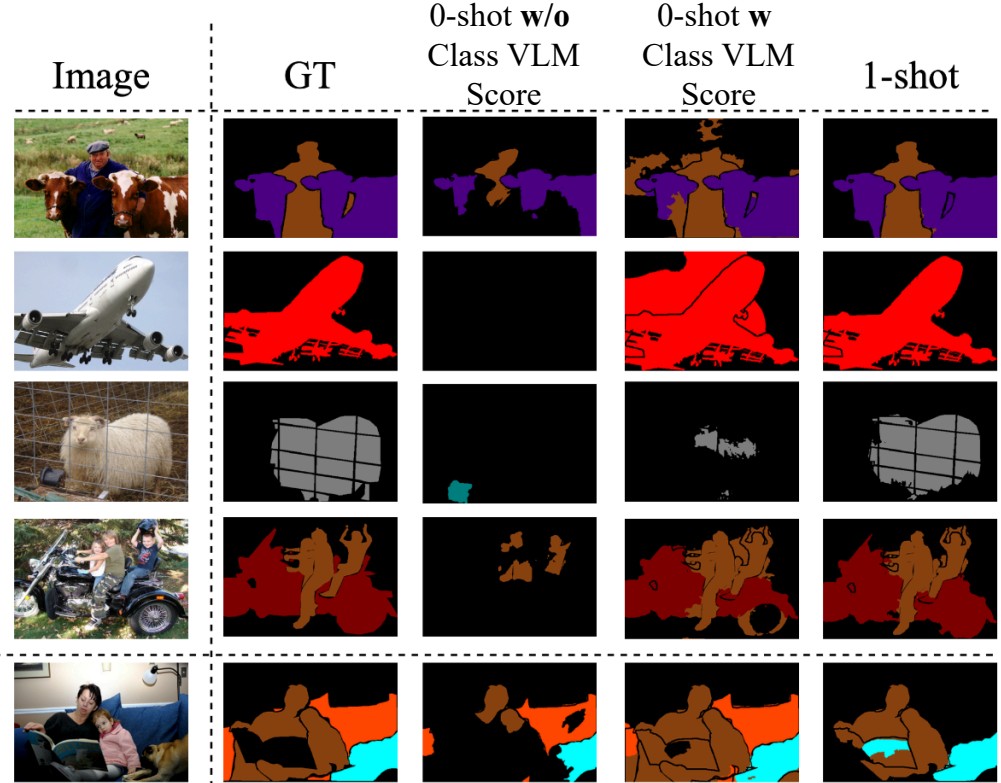

Figure 7: **Qualitative Results on PASCAL-21:** Shown are results from zero-shot training-free w/o image-text scoring (3rd column), zero-shot model w/ image-text scoring (4th column), and one-shot fine-tuning (5th column). The final row shows an example where the zero-shot prediction outperformed the fine-tuned one-shot model. For all variants, we ensemble the top-2 layers from BLIP ranked by InfoScore. All the variants use **multi-prompt per class setting**.

### 4.5 Parameter Selection for One-Shot Fine-Tuning

Table 10 demonstrates the effectiveness of our selective fine-tuning strategy, where we fine-tune only the word embeddings and the top-2 image-text cross-attention layers in BLIP. We compare six fine-tuning variants: (i) fine-tuning only the top-2 cross-attention layers selected by InfoScore, (ii) fine-tuning only the word embeddings, (iii) fine-tuning both the word embeddings and two randomly picked cross attention layers (over 5 runs), (iv) fine-tuning both the word embeddings and the top-2 cross-attention layers, (v) fine-tuning all the cross-attention layers alongwith word embeddings, and (v) fine-tuning all VLM paramters. The results show that fine-tuning both the word embeddings and the top-2 cross-attention layers selected by InfoScore consistently yields the best performance. Notably, increasing the number of tunable parameters beyond this point leads to reduced performance, highlighting the effectiveness of our selective fine-tuning.

Overall, BLIP consists of approximately 446.13 million parameters. In our selective fine-tuning strategy, we tune only the top-2 cross-attention layers and the word embeddings corresponding to the prompts. Although BLIP's vocabulary size is 30,524 words, in practice, we fine-tune only the embeddings of the words that appear in the prompts—typically up to around 250 word embeddings. This results in fine-tuning approximately 5.7 million parameters out of 446.13 million, which is just 1.28% of the total parameters. Importantly, we do not introduce any additional parameters in either the training-free or few-shot fine-tuning settings. Our method

leverages the existing VLM architecture to generate segmentation maps efficiently, without increasing model size, demonstrating both scalability and parameter efficiency.

## 4.6 Qualitative Results

Figure 7 presents the qualitative results of our approach across different settings, illustrating the impact of image-text scoring and one-shot fine-tuning. As shown, the model without image-text scoring consistently under-segments across all images, labeling most of the pixels as background. Applying image-text scoring for filtering notably improves performance across all images, as it helps the model better focus on relevant areas. However, there is a tendency to over-segment certain categories, as seen in the first couple of rows. One-shot fine-tuning further enhances segmentation accuracy, consistently producing more precise segmentations. However, the last row highlights a failure case for the one-shot fine-tuning, where potential bias from the single example of the class `couch` led to mis-classification.

We provide additional qualitative results on the **COCO-Obj** and **ADE-20K** datasets in Figures 10 and 11, respectively, with further discussion in Section A.4 of the Appendix.

## 5 Limitations

Our method is applicable to a wide range of modern vision–language models that expose explicit multimodal interactions between image and text tokens. Its effectiveness depends on the quality of the resulting multimodal attention maps. In particular, successful segmentation requires attention maps that exhibit strong text–image grounding and sufficient spatial resolution. Architectures that aggressively merge or downsample visual tokens may produce attention maps that are too coarse for fine-grained pixel-level segmentation.

In addition, our framework is not applicable to CLIP-like encoder-only vision–language models that compute image and text representations independently and do not expose multimodal attention during inference. Such models fall outside the scope of this work.

## 6 Conclusion

Our work extends state-of-the-art approaches that perform segmentation by using vision foundation models by eliminating the need for abundant segmentation labels. We leverage the strength of a single visual example to better disambiguate categories beyond their textual names. This, combined with our proposed InfoScore measure, reduces reliance on intensive prompt engineering or tuning of the layers/heads selected for segmentation in VLMs. Our approach can operate in both training-free and one-shot fine-tuning settings, with the latter achieving significant gains on four benchmarks and demonstrating compatibility across three different VLMs.

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

# A  Appendix

## A.1  Further Implementation Details

For BLIP (Li et al., 2022b) and LLaVA (Liu et al., 2024a), we rely on the `HuggingFace`[2] library. For ALBEF (Li et al., 2021), we use `LAVIS`[3].

---

[2]`https://huggingface.co/docs/transformers/en/model_doc/blip`
[3]`https://github.com/salesforce/LAVIS`

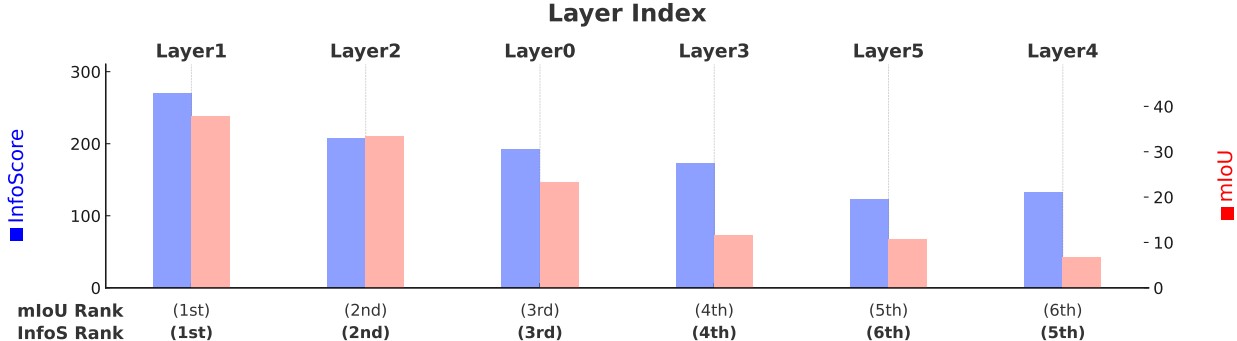

Figure 8: **Illustration of the InfoScore measure on ALBEF.** The mIoU Rank reflects the descending order of mIoU values (in red) derived from cross-attention maps for each standalone layer (labeled Layer$N$, top) on the PASCAL VOC 2012 validation set (1449 images), compared to the predicted InfoS Rank (bottom) based on our InfoScore measure (in blue) requiring no annotations. InfoScore rankings align with mIoU Rankings except for the worst two layers with rankings interchanged between 5th and 6th. Results shown here are in **single-prompt per class setting**.

**BLIP.** For BLIP we use pretrained model weights of `Salesforce/blip-itm-large-coco`. It uses ViT-L/16 backbone pre-trained on ImageNet as vision encoder and BERT (Devlin et al., 2019) as image-grounded text encoder. The cross-attention layers of BERT take image embeddings as key, value. There are 12 such cross-attention layers and the number of heads is 12. The BLIP model is trained for image-text understanding using a combination of Image-Text Matching (ITM) and Image-Text Contrastive (ITC) Loss on 129M image-text pairs The ITM head of BLIP model is used for class-scoring.

**ALBEF.** For ALBEF we use `albef-retrieval-coco` checkpoint as pre-trained model weights. It uses ViT-B/16 pretrained on ImageNet. Similar to BLIP they are also trained with ITM and ITC Loss but on a much smaller number of image-text pairs of 14.1M. It has six multi-modal layers where there is explicit cross-attention between image and text.

**LLaVA.** For LLaVA we use `llava-1.5-7b-hf` as pre-trained model weights. It uses CLIP VIT-L/14 as a vision encoder which is pre-trained with 400M image-text pairs. The image embeddings are first passed into a projection layer which are then passed to a large language model relying on Vicuna-7B.

**One-shot fine tuning details.** For COCO-Obj, PASCAL-21 and ADE-20K we use training images cropped to maximum width of 512, while for COCO-171 we use a resolution of 256 for training but 512 during inference time. In order to fine-tune LLaVA efficiently, we employ Low-Rank Adapters (LoRA) (Hu et al., 2022) with 4-bit quantization (Dettmers et al., 2023), and selectively fine-tune the LoRA adapters only in the layers identified by InfoScore, along with the word embeddings.

| PostProc | Inference Time | PASCAL-21 | COCO-Obj |
|---|---|---|---|
| ConvCRF (Teichmann & Cipolla, 2018) | **0.4s** | **60.2** | **42.8** |
| PAMR (Araslanov & Roth, 2020) | 3.6s | 59.6 | 42.2 |

Table 11: **Ablation with two different post-processing methods in the training-free setting.** ConvCRF (Teichmann & Cipolla, 2018) marginally outperforms PAMR (Araslanov & Roth, 2020) while being significantly faster at inference. Even with PAMR post-processing, our training-free setting outperforms state-of-the-art training-free approaches like NACLIP (Hajimiri et al., 2025) on COCO-Obj by nearly 6%. Results are in **multi-prompt per class setting**. Best results are in **bold**.

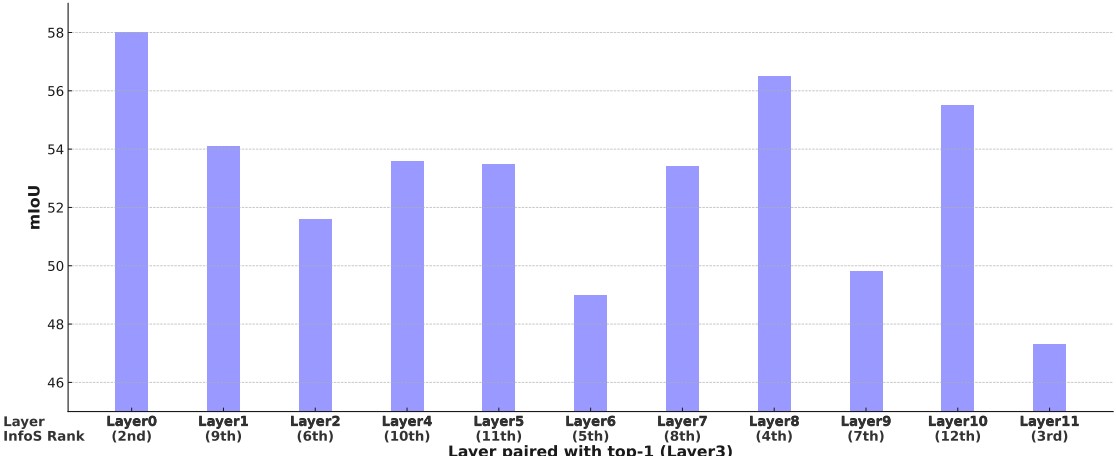

Figure 9: **Performance analysis of pairing layers with the top-1 layer.** We compare the mIoU achieved by pairing the top-1 layer (Layer 3) with each remaining layer using BLIP, evaluated on Pascal-21.

## A.2 InfoScore Additional Analysis

**Layerwise analysis of mIoU and InfoScore on ALBEF.** Following the analysis conducted for BLIP (see section 4.2.3, we perform a layerwise study on ALBEF by independently evaluating each layer for training-free segmentation on the Pascal-21 validation set under the **single-prompt per class** setting. As illustrated in Figure 8, segmentation accuracy exhibits noticeable variation across layers.

We further compute the corresponding InfoScore measure for each layer and compare the induced ranking against the oracle mIoU-based ranking. We observe that InfoScore correctly identifies the top-4 layers, with only a minor swap in the ordering of the last two layers. This close alignment with oracle rankings demonstrates that InfoScore reliably captures layer informativeness and grounding quality, consistent with our observations on BLIP.

**Performance of top-1 paired with others on BLIP.** We ablate a challenging setting where we use the top-1 layer of BLIP identified with our InfoScore measure paired with all the other layers in BLIP evaluated on PASCAL-21 dataset. Figure 9 shows that pairing the top-1 layer (Layer3) with the second best layer (Layer0), outperforms all the other pairs. Thus, it confirms the benefits from our proposed approach even in the challenging setting.

## A.3 Choice of Post-processing Algorithm

In this section, we compare post-processing with ConvCRF (Teichmann & Cipolla, 2018) and PAMR (Araslanov & Roth, 2020), used by recent methods (Wang et al., 2024a; Hajimiri et al., 2025). Table 11 shows that our approach is robust to the choice of post-processing, with ConvCRF providing a slight improvement and being nine times faster than PAMR. In the zero-shot setting on COCO-Obj, our method outperforms the second-best approach, NACLIP (Hajimiri et al., 2025), by 6%, regardless of post-processing. These results confirm that our approach consistently outperforms state-of-the-art methods, independent of the post-processing strategy.

## A.4 Additional Qualitative Results

### A.4.1 Qualitative Results on COCO-Obj

We show additional qualitative results on COCO-Obj dataset in Fig. 10. It shows eight successful scenarios, where the variant without class VLM Score based re-weighting either under-segments or mis-classifies the objects in the scene. On the other hand, the one-shot variant with only a single visual example shows

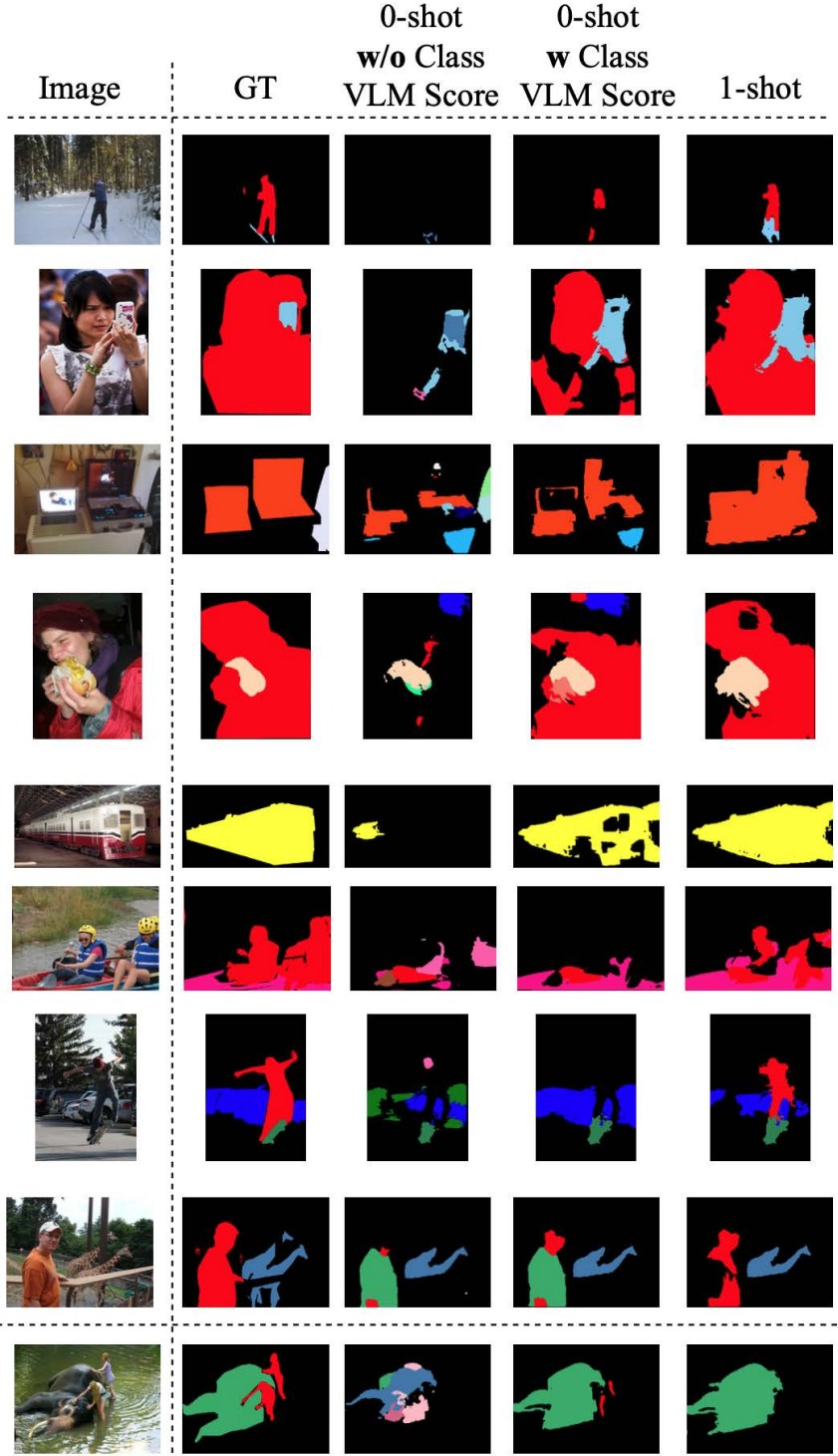

Figure 10: **Qualitative Results on COCO-Obj:** Shown are results from zero-shot model w/o image-text scoring for filtering (3rd column), zero-shot model w/ image-text scoring (4th column), and one-shot fine-tuning (5th column). The final row shows an example where the zero-shot prediction outperformed the fine-tuned one-shot model. For all variants, we ensemble the top-2 layers ranked by InfoScore. All variants use **multi-prompt per class setting**.

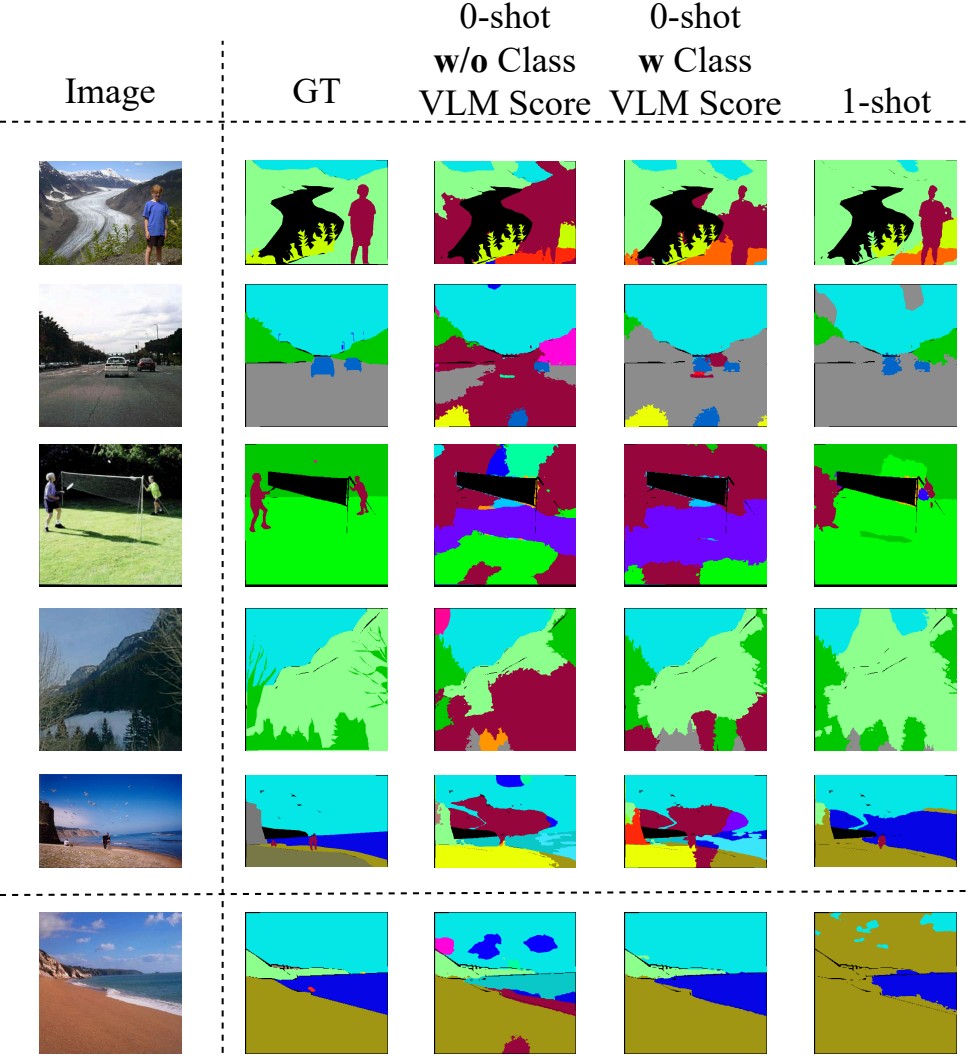

Figure 11: **Qualitative Results on ADE-20K:** Shown are results from zero-shot model w/o image-text scoring for filtering (3rd column), zero-shot model w/ image-text scoring (4th column), and one-shot fine-tuning (5th column). The final row shows an example where the zero-shot prediction outperformed the fine-tuned one-shot model. For all variants, we ensemble the top-2 layers ranked by InfoScore. All variants use **multi-prompt per class setting**.

considerable improvement in segmenting the objects in the scene and overcomes the aforementioned issues. Looking specifically at the second and seventh rows we see that the one-shot variant resolved the confusion on whether to segment the clothes as part of class `Person` or not. Since by definition of the word itself it might exclude the clothes as another class, but with the class definition of COCO-Obj these are to be considered part of `Person` class.

### A.4.2 Qualitative Results on ADE-20K

We present qualitative results on ADE-20K in Fig. 11, comparing zero-shot inference without class VLM score–based re-weighting, zero-shot inference with class VLM score–based re-weighting, and one-shot fine-tuning. As observed in general, the zero-shot variant without class VLM score often suffers from fragmented regions, under-segmentation, or incorrect semantic assignments. Incorporating class VLM scores consistently improves region coherence and semantic alignment, reducing spurious predictions across diverse scenes. The one-shot variant further refines object boundaries and resolves remaining ambiguities, producing segmentations that more closely align with the ground truth.

For example, in the first row, the zero-shot variant without class conditioning produces fragmented and noisy regions, while class-aware scoring substantially improves semantic coherence; the one-shot model further sharpens object boundaries. In the third row, class-aware scoring helps suppress spurious background predictions around the foreground structure, which are further refined after one-shot adaptation.

Finally, the last row highlights a case where zero-shot inference with class-aware scoring outperforms one-shot fine-tuning, indicating that strong class-conditioned priors can already yield competitive segmentations without adaptation.

### A.5 List of Prompts

In this section, we provide the detailed prompts used in the multiple-prompt setting. Our prompt format is `[Image of {class}.]`, where `{class}` represents the class name in the single-prompt setting. For the single-prompt-per-category setting, we made minor modifications to the names of certain categories. For example, in COCO-Obj, we corrected the misspelling of `hair drier` to `hair dryer`. In COCO-171, we removed ambiguous suffixes (*e.g.*, `-other, -stuff`) and renamed classes like `floor-wood` to `wooden floor`. In the multiple-prompt variant, we additionally used synonyms, hyponyms, and/or plurals of the class names.

Table 13 shows the prompts used for PASCAL-21 in the multiple-prompt setting, while Table 12 presents the prompts used for COCO-Obj (80 classes) and COCO-171 (171 classes). The table includes both the 'things' classes, which are common in both datasets, and the 'stuff' classes, which are exclusive to COCO-171.

Note that the list of classes for the background is adopted from the implementations of SCLIP (Wang et al., 2024a)[4] and NACLIP (Hajimiri et al., 2025)[5], and is used consistently for both the single and multi-prompt settings. For background class aggregation, in cross-attention-based models like BLIP and ALBEF, we sum the predictions of all background classes, whereas for LLM-based models like LLaVA, we simply take the maximum prediction among the background classes.

---

[4]https://github.com/wangf3014/SCLIP/blob/main/configs/cls_coco_object.txt
[5]https://github.com/sinahmr/NACLIP/blob/main/configs/cls_coco_object.txt

Table 12: **List of prompts for Single and Multi-Prompt Per Class Settings for COCO-Obj (Things Classes) and COCO-171 (Things + Stuff)**. The symbol * for 'background' indicates that additional prompts are used to represent the background classes for the single prompt setting, following (Wang et al., 2024a; Hajimiri et al., 2025). The list of classes for the background is borrowed from the implementation of SCLIP (Wang et al., 2024a) and NACLIP (Hajimiri et al., 2025). Our prompt takes the form `[Image of {class}.]`, where the {class} is given below. It is to be noted that COCO-171 does not have any background class and the background prompts are used for COCO-Obj only.

| | Class Id and Prompt | Additional Prompts (Multi-prompt) |
|---|---|---|
| Background | 0: background* | sky, wall, tree, wood, grass, road, sea, river, mountain, sands, desk, building, cloud, lamp, door, window, wardrobe, ceiling, shelf, curtain, stair, floor, hill, rail, fence |
| Things Classes | 1: person | people, man, woman, child, children, boy, girl |
| | 2: bicycle | bicycles, bike |
| | 3: car | cars. |
| | 4: motorcycle | motorcycles, motorbike |
| | 5: airplane | airplanes, aeroplane, aircraft |
| | 6: bus | buses, coach |
| | 7: train | - |
| | 8: truck | trucks, lorry |
| | 9: boat | ship, boats, yacht, sailboat, speedboat |
| | 10: traffic light | - |
| | 11: fire hydrant | - |
| | 12: stop sign | - |
| | 13: parking meter | - |
| | 14: bench | benches |
| | 15: bird | birds |
| | 16: cat | cats, kitten |
| | 17: dog | dogs, puppy |
| | 18: horse | horses |
| | 19: sheep | - |
| | 20: cow | cows, cattle |
| | 21: elephant | elephants |
| | 22: bear | bears |
| | 23: zebra | zebras |
| | 24: giraffe | giraffes |
| | 25: backpack | backpacks |
| | 26: umbrella | parasol, umbrellas |
| | 27: handbag | - |
| | 28: tie | necktie |
| | 29: suitcase | - |
| | 30: frisbee | - |
| | 31: skis | ski |
| | 32: snowboard | snowboards |
| | 33: sports ball | ball, sports balls |

| | Class Id and Prompt | Additional Prompts (Multi-prompt) |
|---|---|---|
| Things Classes | **34:** kite | kites |
| | **35:** baseball bat | - |
| | **36:** baseball glove | - |
| | **37:** skateboard | skateboards |
| | **38:** surfboard | surfboards |
| | **39:** tennis racket | racket, tennis rackets, racquet |
| | **40:** bottle | bottles |
| | **41:** wine glass | - |
| | **42:** cup | cups |
| | **43:** fork | forks |
| | **44:** knife | knives |
| | **45:** spoon | - |
| | **46:** bowl | dish |
| | **47:** banana | bananas |
| | **48:** apple | apples |
| | **49:** sandwich | sandwiches |
| | **50:** orange | oranges |
| | **51:** broccoli | - |
| | **52:** carrot | carrots |
| | **53:** hotdog | hotdogs, sausage |
| | **54:** pizza | - |
| | **55:** donut | donuts |
| | **56:** cake | cakes |
| | **57:** chair | chairs |
| | **58:** couch | sofa |
| | **59:** potted plant | indoor plant |
| | **60:** bed | - |
| | **61:** dining table | - |
| | **62:** toilet | - |
| | **63:** tv | television, tvs, television screen |
| | **64:** laptop | - |
| | **65:** mouse | computer mouse |
| | **66:** remote | remote control, remotes |
| | **67:** keyboard | - |
| | **68:** cell phone | cell phones, mobile phone |
| | **69:** microwave | - |
| | **70:** oven | - |
| | **71:** toaster | - |
| | **72:** sink | - |
| | **73:** refrigerator | fridge |
| | **74:** book | books |
| | **75:** clock | - |
| | **76:** vase | - |
| | **77:** scissors | - |
| | **78:** teddy bear | teddy |
| | **79:** hair dryer | blow dryer |
| | **80:** toothbrush | - |
| | **81:** banner | - |
| | **82:** blanket | - |
| | **83:** branch | branches, tree branch |
| | **84:** bridge | - |
| | **85:** building | buildings |

| | Class Id and Prompt | Additional Prompts (Multi-prompt) |
|---|---|---|
| | **86:** bush | bushes |
| | **87:** cabinet | storage, wall cabinet |
| | **88:** cage | - |
| | **89:** cardboard | - |
| | **90:** carpet | - |
| | **91:** ceiling | - |
| | **92:** tile ceiling | - |
| | **93:** cloth | - |
| | **94:** clothes | - |
| | **95:** clouds | - |
| | **96:** counter | - |
| | **97:** cupboard | - |
| | **98:** curtain | - |
| | **99:** desk | - |
| | **100:** dirt | - |
| | **101:** door | - |
| | **102:** fence | - |
| | **103:** marble floor | - |
| | **104:** floor | - |
| | **105:** stone floor | - |
| | **106:** tiled floor | - |
| | **107:** wooden floor | - |
| | **108:** flower | - |
| | **109:** fog | - |
| | **110:** food | - |
| | **111:** fruit | - |
| | **112:** furniture | - |
| | **113:** grass | - |
| | **114:** gravel | - |
| Stuffs Classes | **115:** ground | - |
| | **116:** hill | - |
| | **117:** house | - |
| | **118:** leaves | - |
| | **119:** light | - |
| | **120:** mat | door mat |
| | **121:** metal | metal surface, metallic object |
| | **122:** mirror | - |
| | **123:** moss | spores, mosses |
| | **124:** mountain | - |
| | **125:** mud | - |
| | **126:** napkin | - |
| | **127:** net | - |
| | **128:** paper | - |
| | **129:** pavement | sidewalk, footpath |
| | **130:** pillow | - |
| | **131:** plant | plants |
| | **132:** plastic | - |
| | **133:** platform | - |
| | **134:** playing field | playground |
| | **135:** railing | - |
| | **136:** railroad | - |
| | **137:** river | - |
| | **138:** road | - |

| | Class Id and Prompt | Additional Prompts (Multi-prompt) |
|---|---|---|
| Stuffs Classes | **139:** rock | - |
| | **140:** roof | - |
| | **141:** rug | - |
| | **142:** salad | - |
| | **143:** sand | - |
| | **144:** sea | - |
| | **145:** shelf | - |
| | **146:** sky | - |
| | **147:** skyscaper | - |
| | **148:** snow | - |
| | **149:** solid material | - |
| | **150:** stairs | - |
| | **151:** stone | - |
| | **152:** straw | - |
| | **153:** structure | - |
| | **154:** table | - |
| | **155:** tent | - |
| | **156:** textile | - |
| | **157:** towel | - |
| | **158:** tree | - |
| | **159:** vegetable | - |
| | **160:** brick wall | - |
| | **161:** concrete wall | - |
| | **162:** wall | - |
| | **163:** wall panel | - |
| | **164:** stone wall | - |
| | **165:** tiled wall | - |
| | **166:** wooden wall | - |
| | **167:** water | - |
| | **168:** drops of water | - |
| | **169:** window blind | - |
| | **170:** window | - |
| | **171:** wood | timber |

| Class Id and Prompt | Additional Prompts |
|---|---|
| **0:** background* | sky, wall, tree, wood, grass, road, sea, river, mountain, sands, desk, building, cloud, lamp, door, window, wardrobe, ceiling, shelf, curtain, stair, floor, hill, rail, fence |
| **1:** airplane | aeroplane, jet, airplanes, plane, aeroplanes, jets, planes |
| **2:** bicycle | bicycles, bike, bikes |
| **3:** bird | birds |
| **4:** boat | boats, yacht, ship, ships, speedboat, speedboats, yachts |
| **5:** bottle | bottles |
| **6:** bus | buses, coach, coaches |
| **7:** car | cars |
| **8:** cat | cats |
| **9:** chair | chairs, dining chair |
| **10:** cow | cows, cattle |
| **11:** dining table | dining tables |
| **12:** dog | dogs |
| **13:** horse | horses |
| **14:** motorcycle | motorcycles, motorbike, motorbikes |
| **15:** person | people, man, woman, men, women, boys, girls, child, children, boy, person in shirt, person in jeans, person in dress, person in sweater, person in skirt, person in jacket |
| **16:** potted plant | potted plants, indoor plants, house plants |
| **17:** sheep | - |
| **18:** couch | sofa, couches |
| **19:** train | trains, railcar, railcars |
| **20:** tv | television, television set, television monitor, tv monitor, monitor, television screen, TVs |

Table 13: **List of prompts for Single and Multi-Prompt Per Class Settings for PASCAL-21.** The symbol * for 'background' indicates that additional prompts are used to represent the background classes for the single-prompt setting, following (Wang et al., 2024a; Hajimiri et al., 2025). The list of background classes is borrowed from the implementations of SCLIP (Wang et al., 2024a) and NACLIP (Hajimiri et al., 2025). Our prompt format is [Image of {class}.], where {class} is as listed above.

