# OpenReview forum: "Segmentation From Attention: Training-Free Layer Selection and One-Shot Tuning for Segmentation in VLMs"
_TMLR — Accepted by TMLR_

### Review · Reviewer_tJRW · 2025-11-23

**Summary Of Contributions:**

In this paper, the authors study the problem of obtaining training-free segmentation maps from attention layers in vision-language models (VLMs). To this end, the authors (i) introduce an entropy-based measure, called InfoScore, that can quantify the relevance of an attention layer for a class and (ii) propose selecting most informative layers (either in a training free fashion or for one-shot training) through their InfoScore measure. Experiments on several datasets suggest improvements over the baselines.

Strengths:

+ Training-free acquisition of segmentation maps is an important research direction in computer vision.
+ The paper proposes a novel measure for quantifying the relevance of attention layers in a VLM.
+ Positive results on several benchmarks.


Key Weaknesses:
- Some assumptions are not justified.
- Experimental evaluation needs to be more comprehensive.

**Audience:**

Yes

**Audience Explanation:**

TMLR appears to have segmentation papers published before.

**Broader Impact Concerns:**

No concerns.

**Claims And Evidence:**

No

**Claims Explanation:**

6. Experimental evaluation should be strengthened:

6.1. Comparison to other few-shot/one-shot methods is missing.

6.2. Fig 4 caption: "The metric nDCG@k (normalized discounted cumulative gain) measures the quality of the ranking for the top-k layers, where a perfect ranking compared to the groundtruth ordering yields a score of 1.0." => How do you know the ground truth ranking of the layers? This is not explained in the text either.

6.3. The comparison with NACLIP is not fair since NACLIP uses CLIP whereas the proposed methods uses BLIP. It is important that the proposed method is evaluated and compared with other methods using CLIP as well.

6.4. Table 4: How did you choose these VLMs?

**Requested Changes:**

Weaknesses:

1. The paper's methodology is difficult to follow owing to mostly verbal and informal (notation-less) explanations of the steps.

2. Some comparisons or arguments are not substantiated:

2.1. Figure 1 (right):

2.1.1. What is the performance of existing layer selection methods? The comparison to random selection and including all layers shows an incomplete picture.
2.1.2. What would be the performance of random selection and including all layers with one-shot tuning?

3. It would be helpful to add a related work section on measuring segmentation map properties.

4. The properties of the new measure should be justified by introducing desiderata for the new measure.

5. The new measure is called a metric. Metricity requires satisfying certain properties. Please provide that these properties are satisfied or call it a measure.

6. Experimental evaluation should be strengthened:

6.1. Comparison to other few-shot/one-shot methods is missing.

6.2. Fig 4 caption: "The metric nDCG@k (normalized discounted cumulative gain) measures the quality of the ranking for the top-k layers, where a perfect ranking compared to the groundtruth ordering yields a score of 1.0." => How do you know the ground truth ranking of the layers? This is not explained in the text either.

6.3. The comparison with NACLIP is not fair since NACLIP uses CLIP whereas the proposed methods uses BLIP. It is important that the proposed method is evaluated and compared with other methods using CLIP as well.

6.4. Table 4: How did you choose these VLMs?


Minor comments:
- "on foundation models Bommasani et al. (2021); Li et al. (2024)" => If a citation is a part of the sentence (e.g., "Bommasani et al. (2021) proposed a novel method.."), then the citation should be included as you did. However, if the citation is not a part of the sentence, it should stay within parentheses. The correct way to write the selected text is: "on foundation models (Bommasani et al., 2021; Li et al., 2024)". One of the cite commands in Latex handles this correctly. Try cite, citep or citet.
- Figure 1 (right): Please export/save your plots as scalable graphics, e.g. SVG, PDF, and include them as such in Latex.
- Section 3.1: Start with the first paragraph, I would suggest introducing symbols for tensors and modules/models, e.g., query image, prompt, layer, attention score, words, class, ... In its current state, the method is very difficult to follow without proper notation.
- Fig 2: Font sizes can be increased.
- "the probability that the pixel (x, y) belongs to class c" => Previously you used x to represent a pixel and y to represent a label. Please make the notation more coherent.
- Eq 4: The summations run over i and j but the subscripts run over x and y.
- Eq 10: So, this reduces to \sum_c sigma_c/mu_c, right?

---

> ### Author Response · Authors · 2025-12-15
> **Response to Reviewer tJRW**
>
> ### Response to Reviewer tJRW
>
> We thank the reviewer for the time and effort invested in reviewing the manuscript and for the detailed, constructive feedback, which helped us substantially improve the paper.
>
> ### Methodology clarity and formalization:
>
> To improve clarity and rigor, we have substantially revised **Section 3.1** to use more formal notation and structured explanations. In particular, we have added two pseudocodes: **Algorithm 1**, which details the computation of InfoScore, and **Algorithm 2**, which describes the zero-shot inference procedure. Please refer to the revised manuscript.
>
> ### Regarding existing layer selection approaches and comparison to random layers and all layers showing an incomplete picture:
>
> Existing approaches for grounding and segmentation (e.g., Luo et al., 2024; He et al., 2024; Li et al., 2021) typically rely on some form of supervision for layer selection, either by selecting layers or heads based on performance on labeled validation data, or by providing the names of classes present in the image and using an external model—i.e., requiring some form of ground-truth information. Consequently, the strongest possible baseline in this setting is an oracle selection that ranks layers using ground-truth mIoU.
>
> To contextualize our results, we already include oracle rankings based on ground-truth mIoU in Figure 5 and Table 8 of the original submission. As shown there, InfoScore closely matches this oracle ordering, experimentally recovering the optimal layer ranking without access to labels or pseudo masks. These results demonstrate that InfoScore achieves performance comparable to the best-case scenario among existing supervised or weakly supervised layer-selection strategies.
>
> ### One-shot performance of random selection and all layers:
>
> We have extended **Figure 1** and **Table 9** in the revised manuscript to include one-shot results for random layer selection and selecting all layers. These additional results show that while one-shot tuning reduces the performance gap between different layer-selection strategies, InfoScore-based selection still consistently outperforms random selection and selecting all layers. This confirms that even in the presence of limited supervision, informed layer selection provides tangible benefits.
>
> ### Related work on measuring segmentation map properties:
>
> We appreciate the concern raised by the reviewer. However, we would appreciate clarification on what is meant by measuring segmentation map properties, so that we can address the concern appropriately. Our work focuses on selecting informative attention layers by analyzing statistical properties of attention-derived class probability maps, rather than defining or evaluating properties of final segmentation masks.
>
> If the reviewer could clarify the specific line of prior work they are referring to, we would be happy to address this suggestion more precisely in the revision.
>
> ### Design desiderata for InfoScore:
>
> We thank the reviewer for this suggestion. The desiderata for the proposed measure are already discussed in **Section 3.2**, where we motivate InfoScore by requiring:
>
> - confident predictions within each image,
> - diverse predictions across the dataset, and
> - meaningful image-to-image variation.
>
> These principles directly motivate the three components of InfoScore: mean image-level entropy, dataset-level entropy, and the coefficient of variation.
>
> To address the reviewer’s concern, we have revised **Section 3.2** to explicitly enumerate these desiderata before introducing the formal definition of InfoScore, making the design rationale clearer.
>
> ### Calling InfoScore a measure instead of a metric:
>
> We thank the reviewer for pointing this out. We have revised the manuscript to refer to InfoScore as a *measure* rather than a *metric*, and updated the text accordingly in the revised manuscript.
>
> ### Comparison to other one-shot methods:
>
> We thank the reviewer for the suggestion. As requested, we have added 1-shot evaluations following the standard few-shot segmentation protocol. The results are now reported in **Table 2** and discussed in **Section 4.2.2** of the revised manuscript. As shown, our method achieves competitive performance without any base-class pre-training and attains state-of-the-art results when evaluated under the standard few-shot protocol with base-class training. Further details are provided in **Section 4.2.2**.

---

> > ### Author Response · Authors · 2025-12-15
> > **Continuation of response to Reviewer tJRW**
> >
> > ### How ground-truth ranking was used for nDCG@k:
> >
> > The ground-truth layer ordering used to compute nDCG@k is obtained via an oracle ranking. Specifically, layers are ranked according to their actual segmentation performance (mIoU) on a labeled validation set, with higher mIoU indicating a better layer. This oracle ranking is used only for evaluation of the ranking quality and is not available at inference time.
> >
> > Importantly, InfoScore itself is computed in a fully unsupervised and training-free manner and does not use any labels or annotations.
> >
> > ### Comparison with NACLIP not fair because it uses CLIP while ours uses BLIP:
> >
> > As shown in **Table 1** of the main paper, our evaluation is not limited to NACLIP nor to a single type of VLM. We compare against a broad set of state-of-the-art open-vocabulary segmentation methods built on different vision–language backbones, including multiple **CLIP-based approaches** (e.g., NACLIP, SCLIP, GroupViT), **BLIP-based methods** (e.g., PnP-OVSS), **MetaCLIP-based methods** (e.g., Grounding Everything), and hybrid approaches that **combine CLIP with additional vision encoders** (e.g., ProxyCLIP). All of these methods operate in a **training-free open-vocabulary setting** using models that are not trained with segmentation annotations, which makes them appropriate and fair baselines for comparison.
> >
> > Moreover, the proposed framework is applicable to a wide range of modern vision–language models that perform explicit multimodal interaction between image and text tokens, including **cross-attention–based models** (e.g., BLIP, ALBEF) and **multimodal self-attention–based models** (e.g., LLaVA). CLIP-style two-tower encoders follow a different architectural design and fall outside the scope of this work, as already clarified in our response to reviewer guZd and also discussed in **Section 5** of the revised manuscript.
> >
> > ### How we selected VLMs:
> >
> > The models included in **Table 4** were chosen to support our goal of developing an approach that is applicable to a broad family of vision–language models with explicit multimodal interactions. In particular, we focus on architectures that expose interpretable interactions between image and text tokens, either through cross-attention (e.g., BLIP, ALBEF) or multimodal self-attention (e.g., LLaVA), which is a prerequisite for extracting spatially meaningful attention maps.
> >
> > In addition to architectural considerations, we prioritize models for which prior work has reported strong grounding behavior (*Luo et al., 2024; He et al., 2024*), indicating that textual concepts are well aligned with spatial image regions. This property is essential for attention-based localization and segmentation, and helps ensure that our evaluation focuses on settings where multimodal attention is both available and meaningful. Together, these considerations guided our choice of VLMs and ensure coverage across diverse multimodal interaction paradigms.
> >
> > ### Minor comments regarding citations, stylistic changes, and equations:
> >
> > Addressed all minor comments, including citation formatting, notation consistency, equation corrections, figure readability, and the use of scalable vector graphics for plots.

---

> > ### Comment · Reviewer_tJRW · 2025-12-17
> >
> > I would like to thank the authors for their detailed responses, additional experiments and results. I strongly believe that the paper has improved significantly with this revision.
> >
> > My remaining concerns and suggestions:
> >
> > * Can you please include the oracle results in Figure 1 (right)? The left subfigure would benefit from enlarging the font size (something that matches the font size of the right subfigure). It is not a must be a suggestion: I would place sub-labels under the left and right subfigures as (a) and (b).
> >
> > * Related work on segmentation map properties: Sorry, I meant "heatmap / attention-map properties". The literature has used information-theoretical measures and other methods to quantify different characteristics of heatmaps/saliency-maps. I believe a discussion of such approaches is helpful. Some references I could find include:
> >
> > Tian, Y., Wang, Y., Chen, B., & Du, S. S. (2023). Scan and snap: Understanding training dynamics and token composition in 1-layer transformer. Advances in neural information processing systems, 36, 71911-71947.
> >
> > Petsiuk, V., Das, A., & Saenko, K. (2018). Rise: Randomized input sampling for explanation of black-box models. arXiv preprint arXiv:1806.07421.
> >
> > Pardyl, A., Rypeść, G., Kurzejamski, G., Zieliński, B., & Trzciński, T. (2023). Active visual exploration based on attention-map entropy. arXiv preprint arXiv:2303.06457.
> >
> > * Please use the following guide in writing the equations: https://wp.optics.arizona.edu/kupinski/wp-content/uploads/sites/91/2023/05/MerminEquations.pdf
> >
> > * Fig 3 caption: "InfoScore.We" => "InfoScore. We".
> >
> > * Table 2: What do S<number> denote in the table? Different number of shots? But the table caption and Section 4.2.2 state 1-shot comparisons? What do bold numbers indicate in the table? Best values? Then some outmatched numbers are bolded.

---

> > > ### Author Response · Authors · 2025-12-18
> > > **Response to the official Comment by Reviewer tJRW**
> > >
> > > ### Response to the official Comment by Reviewer tJRW
> > >
> > > We thank the reviewer for the encouraging assessment of the revision and are pleased to hear that the additional experiments and clarifications improved the manuscript. We appreciate the constructive suggestions and address the remaining points below. We hope these responses adequately address the reviewer’s remaining concerns.
> > >
> > > ---
> > >
> > > ### Figure 1 (oracle results, font size, sub-labels)
> > > We will incorporate all of the suggested updates, including adding oracle results for both the 0-shot and 1-shot settings, increasing font sizes, and adding sub-labels, in the camera-ready version if accepted.
> > >
> > > ---
> > >
> > > ### Formatting and typographical issues
> > > Thank you for pointing us to the equation style guide. We will revise the equation formatting accordingly and will also thoroughly proofread the manuscript to correct typographical issues in the camera-ready version.
> > >
> > > ---
> > >
> > > ### Table 2 clarification
> > > In Table 2, **S0, S1, S2, and S3** denote different predefined splits of base and novel classes, following standard practice in the few-shot segmentation literature. All reported results correspond to **1-shot performance**, evaluated separately on each split and averaged across splits. **Bold numbers indicate the best performance within each split and overall**.
> > >
> > > We acknowledge that one entry is currently bolded inconsistently (on split **S2**, where the variant without base training marginally outperforms the variant with base training), and that the current caption does not make the meaning of the splits and bold formatting sufficiently clear. We will correct the bolding, clarify the caption, and explicitly explain the splits in **Section 4.2.2** in the camera-ready version.
> > >
> > > ---
> > >
> > > ### Related Work on Attention-Map / Heatmap Properties
> > > Thank you for the clarification and references. We agree that situating **InfoScore** within prior work on information-theoretic analysis of heatmaps and attention maps would strengthen the paper. We will add a dedicated paragraph in the camera-ready version; a draft is provided below.
> > >
> > > ---
> > >
> > > ### Quantifying Heatmap and Attention-Map Properties (draft)
> > > A line of work has studied the intrinsic properties of heatmaps and saliency maps, aiming to quantify their informativeness, uncertainty, or structure. Several approaches adopt information-theoretic frameworks, such as entropy, entropy rate, or information gain, to characterize saliency distributions (Wang et al., **CVPR 2010**; Kümmerer et al., **PNAS 2015**) and attention maps (Pardyl et al., **IJCAI 2023**).
> > >
> > > Beyond information-theoretic formulations, prior work has examined other intrinsic properties of saliency maps, including sparsity, dispersion, and calibration, and proposed quantitative metrics for their evaluation (Gomez et al., **ICPRAI 2022**; Gupta et al., **NeurIPS 2022**). A complementary line of research evaluates heatmap quality through perturbation-based and causal criteria, testing whether highlighted regions genuinely influence model predictions. Representative approaches include meaningful and extremal perturbations (Fong & Vedaldi, **CVPR 2017**; Fong et al., **ICCV 2019**), randomized input masking (Petsiuk et al., **arXiv 2018**), and benchmark-based faithfulness and sensitivity analyses (Hooker et al., **NeurIPS 2019**; Yeh et al., **NeurIPS 2019**).
> > >
> > > Other works have examined the reliability of saliency methods, showing that some explanations may be insensitive to model parameters or training data (Adebayo et al., **NeurIPS 2018**; Hedström et al., **XAI 2024**). In transformer-based models, several studies have further questioned whether attention weights can be directly interpreted as explanations, motivating analyses beyond raw attention visualization (Jain & Wallace, **NAACL 2019**; Serrano & Smith, **ACL 2019**; Chefer et al., **CVPR 2021**).
> > >
> > > While these approaches provide valuable tools for analyzing and comparing heatmaps and attention maps, they are not designed to identify informative layers for segmentation. Our work builds on this literature by leveraging information-theoretic properties of attention maps specifically for **training-free layer selection** in VLM-based segmentation.

---

> > > > ### Author Response · Authors · 2025-12-18
> > > > **Continuation of the response to official comment by Reviewer tJRW**
> > > >
> > > > ### Continuation of the response to official comment by Reviewer tJRW
> > > > We have referenced the following works in the draft for related work on Attention-Map / Heatmap properties:
> > > >
> > > > ### Referenced papers
> > > > - Wang et al., 2010 – *Measuring Visual Saliency by Site Entropy Rate*. **CVPR 2010**
> > > > - Kümmerer et al., 2015 – *Information-Theoretic Model Comparison Unifies Saliency Metrics*. **PNAS 2015**
> > > > - Pardyl et al., 2023 – *Active Visual Exploration Based on Attention-Map Entropy*. **IJCAI 2023**
> > > > - Gomez et al., 2022 – *Metrics for Saliency Map Evaluation of Deep Learning Explanation Methods*. **ICPRAI 2022**
> > > > - Gupta et al., 2022 – *New Definitions and Evaluations for Saliency Methods: Staying Intrinsic, Complete and Sound*. **NeurIPS 2022**
> > > > - Fong & Vedaldi, 2017 – *Interpretable Explanations of Black Boxes by Meaningful Perturbations*. **CVPR 2017**
> > > > - Fong et al., 2019 – *Understanding Deep Networks via Extremal Perturbations and Smooth Masks*. **ICCV 2019**
> > > > - Petsiuk et al., 2018 – *RISE: Randomized Input Sampling for Explanation of Black-Box Models*. **arXiv 2018**
> > > > - Hooker et al., 2019 – *A Benchmark for Interpretability Methods in Deep Neural Networks*. **NeurIPS 2019**
> > > > - Yeh et al., 2019 – *On the (In)Fidelity and Sensitivity of Explanations*. **NeurIPS 2019**
> > > > - Adebayo et al., 2018 – *Sanity Checks for Saliency Maps*. **NeurIPS 2018**
> > > > - Hedström et al., 2024 – *A Fresh Look at Sanity Checks for Saliency Maps*. **XAI 2024**
> > > > - Jain & Wallace, 2019 – *Attention Is Not Explanation*. **NAACL 2019**
> > > > - Serrano & Smith, 2019 – *Is Attention Interpretable?* **ACL 2019**
> > > > - Chefer et al., 2021 – *Transformer Interpretability Beyond Attention Visualization*. **CVPR 2021**

---

### Review · Reviewer_ZrbZ · 2025-12-01

**Summary Of Contributions:**

In this paper, the authors propose InfoScore, a training-free entropy-based metric that automatically ranks and selects the most relevant image-text attention layers for open-vocabulary segmentation. The authors introduce a false-positive filtering mechanism that re-weights attention maps using the model's own image-text scores to reduce noise without external supervision. The paper presents a one-shot tuning strategy that selectively fine-tunes the identified attention layers using a single support image, significantly improving performance without adding complex decoders. Experiment results show that the proposed framework is model-agnostic and consistently outperforms naive selection strategies across diverse vision-language model architectures.

**Additional Comments:**

"To compute the overall attention score for a prompt, we take the mean across the words in the prompt, resulting in an attention score of dimension P × P for that prompt. " Why use mean attention score over the prompt here? I don't think this is well justified. Could the authors explain more here? I think using mean attention score is susceptible to the prompt format. For example, one could use "Class *Something*" instead of "*Image of Class 1*".

**Audience:**

Yes

**Audience Explanation:**

I think the research topic and question studied in this paper should be of interest to the TMLR community.

**Broader Impact Concerns:**

I do not have concerns on broader impact of this paper.

**Claims And Evidence:**

Yes

**Claims Explanation:**

- The authors addressed an important topic on how to do zero-shot and few-shot segmentation from vision-language models. Demonstrating that this is possible is an important empirical result.
- In the paper, the models used for experiments are BLIP, LLaVA and ALBEF. These models are not the state-of-the-art vision-language models. Could the authors experiment with more recent multimodal vision-language models, like Qwen2/3-VL or Gemma 3.
- I think analysis like Figure 5 is very important. I think it would be good to provide this analysis for more models and datasets. I am curious if there is any pattern from the selected layers, can this be used to help understand which layers in vision-language models are better at localization/segmentation ability.

**Requested Changes:**

I think the authors need to provide more qualitative analysis in the experiments. Currently in the main paper, only Figure 7 illustrates the qualitative segmentation results. I think this kind of results is more intuitive and can help readers understand the proposed methods better. It would be good to supplement the paper with segmentation results on more diverse image datasets and compare with other state-of-the-art segmentation methods, e.g, SAM-3 [1].

[1] SAM 3: Segment Anything with Concepts. 2025.

---

> ### Author Response · Authors · 2025-12-15
> **Response to Reviewer ZrbZ**
>
> ## Response to Reviewer ZrbZ
>
> We thank the reviewer for the positive assessment of our contributions and for the constructive feedback, which helped us strengthen the revised manuscript. We respond to the concerns raised by the reviewer in detail below.
>
> ### Experiments with More Recent Vision–Language Models (e.g., Qwen2/3-VL, Gemma 3):
>
> We thank the reviewer for the suggestion. Our approach is indeed applicable to modern vision–language models. We provide the the 0-shot evaluation on **QWEN2-VL** on **Pascal-VOC** dataset below:
>
> | Layer Selection Strategy        | mIoU (%) |
> |----------------------------------|----------|
> | InfoScore (Top-2)                   | 32.0     |
> | Naive (First 2 layers)              | 19.4     |
> | Naive (Last 2 layers)              | 27.2     |
> | Random Selection (5 runs)    | 22.1     |
>
> Although InfoScore-based selection continues to outperform naive and random selection strategies, the absolute segmentation performance on Qwen2-VL is lower than for the other VLMs evaluated in the paper. We attribute this primarily to Qwen2-VL’s **2×2 PatchMerger**, which merges visual patches before projecting them into the language space, substantially reducing the number of visual tokens passed to the LLM. This design improves computational and memory efficiency and reduces the dominance of visual tokens for text-generation tasks, but yields coarser attention maps and limits the granularity of the spatial signal available for pixel-level segmentation.
>
> Despite this architectural difference, InfoScore still identifies the most informative layers under this design, consistently outperforming all naive baselines. This provides additional evidence for the robustness and general applicability of our layer-selection strategy across diverse VLM designs.
>
> ### Layerwise Analysis Like Figure 5:
>
> We thank the reviewer for this constructive suggestion. In the revised paper, we added an analysis similar to Figure 5 for additional VLMs. Specifically, **Appendix Figure 8** now includes a full layerwise InfoScore analysis for **ALBEF**, with discussion in **Section A.2**.
>
>
> ### Patterns in Selected Layers among different style of VLMs:
>
> We thank the reviewer for suggesting this interesting analysis. We found that the trend in ALBEF is broadly consistent with BLIP: the most informative multimodal layers tend to appear closer to the visual input, while later layers contribute less useful spatial grounding. This suggests that in cross-attention-based architectures, earlier multimodal fusion layers provide stronger localization signals.
>
> For **LLaVA-1.5-7B**, which uses self-attention-based multimodal integration rather than explicit cross-attention, the pattern differs. Its 32-layer architecture yields the following InfoScore ranking (best layers first):
>
> [14, 21, 23, 19, 26, 24, 28, 15, 18, 17, 25, 13, 20, 22, 10, 29, 30, 9, 8, 11, 27, 16, 31, 6, 7, 5, 12, 4, 3, 2, 1, 0].
>
> Here, the most informative layers appear predominantly in the second half of the network, indicating that self-attention VLMs develop useful spatial grounding deeper into the model than cross-attention architectures.
>
> In addition, we evaluated **Qwen2-VL-7B-Instruct**, which similarly uses self-attention for multimodal interaction. For its 28 layers, the top InfoScore layers are:
>
> [19, 16, 22, 17, 18, 20, 26, 21, 24, 5, 23, 25, 14, 12, 15, 8, 11, 13, 3, 27, 2, 9, 6, 4, 10, 0, 1, 7].
>
> This ranking shows a pattern similar to LLaVA, with the most informative layers again concentrated in the later part of the model.
>
> Together, these analyses suggest a consistent trend:
>
> - **Cross-attention VLMs (BLIP, ALBEF):** more informative layers tend to occur earlier in the multimodal encoder.
> - **Self-attention VLMs (LLaVA, Qwen2-VL):** more informative layers tend to occur later in the network.
>
> ### Additional qualitative results on different datasets:
>
> In the original submission, we provided additional qualitative zero-shot segmentation results on **COCO-Obj**, shown in **Appendix Figure 9** (Figure 10 in the revised version) and discussed in **Section A.4** of the original manuscript (Section A.4.1 in the revised). Following the reviewer’s suggestion, we added qualitative examples on a more diverse dataset, **ADE-20K**, included as **Figure 11** in the revised manuscript and discussed in **Section A.4.2**.

---

> ### Author Response · Authors · 2025-12-15
> **Continuation of response to Reviewer ZrbZ**
>
> ### Comparison with other state-of-the-art segmentation methods (e.g., SAM-3):
>
> We note that SAM-3 is a fully supervised segmentation model trained with large-scale pixel-level annotations, which falls outside the scope of our setting. Our goal is to extract segmentation maps from vision–language models that are not trained for segmentation, using only multimodal attention and optional minimal supervision. Accordingly, we compare against training-free open-vocabulary segmentation methods operating under similar constraints (i.e., models whose backbones are not trained with segmentation annotations), as reported in **Table 1**.
>
> Additionally, as requested by reviewer tJRW, we included **1-shot evaluations** following the standard few-shot segmentation protocol. These results are now reported in **Table 2** in **Section 4.2.2** of the revised manuscript. As shown, our method achieves state-of-the-art performance in both the zero-shot and 1-shot settings. Since our method is not designed to compete with fully supervised segmentation models, a direct comparison to SAM-3 would not be particularly meaningful.
>
> ### Additional Comment: Why Use Mean Attention Over Prompt Tokens?
>
> We agree that prompt formulation can influence attention maps and appreciate the reviewer for pointing this out. In practice, a single word within a prompt can be tokenized into multiple subword tokens, and attention is computed at the token level; therefore, an aggregation step is required to obtain a single attention map per prompt.
>
> Our goal is to perform this aggregation in a parameter-free and model-agnostic manner. Among simple parameter-free choices, mean and max pooling are the most common options. We adopt mean aggregation across all prompt tokens because it provides a stable estimate and avoids over-emphasizing individual tokens, which can occur with max pooling. We have clarified this design choice in the revised manuscript in **Section 3.1**.

---

### Review · Reviewer_guZd · 2025-12-02

**Summary Of Contributions:**

1. Proposal of InfoScore (**Entropy-based Layer Selection**): Unlike existing methods that rely on manual layer selection or ground-truth annotations , the paper proposes InfoScore, an entropy-based unsupervised metric. This metric enables the automatic selection of optimal attention layers in a training-free manner. The authors demonstrate that this approach achieves superior performance compared to naive selection strategies.

2. False-Positive Filtering (**Class VLM Score**) Mechanism: The authors introduce a false-positive filtering mechanism that suppresses noise from classes not present in the image. This method utilizes the VLM's inherent probabilities, such as image-text matching scores or LLM token probabilities, without requiring external models or additional annotations.

3. **Efficient One-shot Fine-tuning Strategy**: The paper proposes an efficient one-shot fine-tuning strategy that fine-tunes only the text embeddings and the top-K attention layers selected by InfoScore. This approach does not introduce additional decoders or parameters. The results show that performance is significantly improved using only a single visual example.

4. **Generalizability across VLM Architectures (Model-Agnostic)**: The authors demonstrate the generalizability of their proposed framework by experimentally verifying its effectiveness across different VLM architectures. This includes both standard cross-attention-based VLMs (such as BLIP and ALBEF) and modern LLM-based VLMs (such as LLaVA).

**Additional Comments:**

Please see the concerns in the above comments.

**Audience:**

Yes

**Audience Explanation:**

Zero-shot segmentation is a field that has been consistently researched, and this paper proposes a simple and accessible method to improve the zero-shot segmentation.

Furthermore, the authors have empirically verified their proposed contributions through extensive experiments.

Therefore, relevant researchers in the community would likely be interested in this paper.

**Claims And Evidence:**

Yes

**Claims Explanation:**

1. Proposal of InfoScore (Entropy-based Layer Selection)
Evidence: The experiments empirically validate that the InfoScore-based selection is superior to other methods. As shown in Table 2 and Table 5, selecting the Top-2 layers using InfoScore consistently outperforms naive strategies (such as selecting the first/last layers) and random selection baselines across multiple datasets and VLM architectures. Furthermore, Figure 5 demonstrates that the layer rankings predicted by InfoScore align closely with the ground-truth mIoU performance, proving its reliability as an unsupervised metric.

2. False-Positive Filtering (Class VLM Score) Mechanism
Evidence: The effectiveness of the proposed filtering mechanism is clearly evidenced in the ablation study presented in Table 7. Removing the Class VLM Score results in a drastic performance drop (e.g., from 58.0% to 25.1% mIoU on PASCAL-21 for BLIP in the training-free setting). Qualitative results in Figure 7 further illustrate that without this scoring, the model suffers from severe under-segmentation by failing to suppress irrelevant classes, confirming the mechanism's critical role in reducing noise.

3. Efficient One-shot Fine-tuning Strategy
Evidence: The efficiency and effectiveness of the one-shot strategy are substantiated by the results in Table 1 and 9. As shown in Table 1, the method achieves a state-of-the-art mIoU of 70.1% on PASCAL-21, significantly outperforming the training-free baseline (60.2%). Additionally, Table 9 confirms that selectively fine-tuning only the top-2 attention layers and word embeddings yields superior performance compared to fine-tuning more parameters, highlighting the method's parameter efficiency.

4. Generalizability across VLM Architectures (Model-Agnostic)
Evidence: The model-agnostic nature of the framework is supported by Table 4, which shows consistent performance gains across diverse architectures, including cross-attention-based models (BLIP, ALBEF) and LLM-based models (LLaVA).

**Concern1**: However, a significant limitation is that the method relies on explicit cross-attention maps (or self-attention with multimodal tokens). This design renders it inapplicable to independent two-tower architectures like **CLIP** (or newer variants like **SigLIP**), which do not utilize cross-attention layers for inference. This limits the framework's ability to leverage the rapid advancements and scalability of these powerful CLIP-style backbones.

**Concern2**: While the paper compares the method against naive baselines (e.g., random selection) on BLIP but not on ALBEF and LLaVA. Extended experiments are needed to verify that the method consistently provides significant gains over naive sampling across all VLMs.

**Requested Changes:**

There are some typos. For example, "off" -> "of" The very last line in Page 11.

---

> ### Author Response · Authors · 2025-12-15
> **Response to Reviewer guZd**
>
> ## Response to Reviewer guZd
>
> We thank the reviewer for their careful reading of the paper and for their kind remarks regarding our contributions, experimental validation, and clarity. We are particularly grateful for the constructive concerns raised, which helped improve the manuscript. Below, we respond to each concern in detail.
>
> ### Applicability to CLIP-style Two-Tower Architectures:
>
> We appreciate the reviewer’s observation. Our method is designed for vision–language models that expose explicit multimodal interactions between image and text tokens, implemented via cross-attention (e.g., BLIP, ALBEF) or multimodal self-attention (e.g., LLaVA). Our goal is to analyze whether such multimodal attention pathways can be used to extract meaningful segmentation maps and, given their variability across layers, how to select the most informative layers in a training-free manner.
> CLIP and SigLIP follow a different encoder-based design and fall outside the scope of this work. A large body of open-vocabulary segmentation methods has been developed already for CLIP using patch–text similarity, and we include such approaches as baselines in our evaluation. We clarify this scope explicitly in **Section 5 (Limitations)** of the revised paper.
>
>
> ### Comparison with Naive Baselines on ALBEF and LLaVA:
>
> The comparison against naive baselines (random selection and selecting all layers) for ALBEF and LLaVA was provided in the **original submission in Table 5 (Table 6 in the revised paper)**, and discussed in **Section 4.3.2**. As shown there, InfoScore-based layer selection, particularly using the Top-2 layers, consistently outperforms both naive strategies and random sampling across all three evaluated VLMs (BLIP, ALBEF, and LLaVA).
>
>
> ### Typographical errors:
>
> Thank you for pointing out the typographical errors. We have carefully proofread the revised manuscript and corrected this and other minor typographical errors.

---

### Decision · Action_Editor_HdZT · 2026-01-05

**Recommendation:** Accept as is

**Audience:**

Yes

**Audience Explanation:**

All the reviewers agree that there is an audience for this work in the TMLR community.

**Claims And Evidence:**

Yes

**Claims Explanation:**

In their final recommendations, all the reviewers acknowledge that the claims are sufficiently supported. The initial concerns of the reviewers, in particular those of tJRW regarding empirical evaluation, have been convincingly addressed by the authors' responses and revisions.